# Three ancient hormonal cues co-ordinate shoot branching in a moss

Yoan Coudert[1†], Wojtek Palubicki[2†], Karin Ljung[3], Ondrej Novak[4,5], Ottoline Leyser[2], C Jill Harrison[1]*

[1]Department of Plant Sciences, University of Cambridge, Cambridge, United Kingdom; [2]Sainsbury Laboratory, University of Cambridge, Cambridge, United Kingdom; [3]Umeå Plant Science Centre, Department of Forest Genetics and Plant Physiology, Umeå University, Umeå, Sweden; [4]Laboratory of Growth Regulators, Centre of the Region Haná for Biotechnological and Agricultural Research, Palacký University and Institute of Experimental Botany ASCR, Olomouc, Czech Republic; [5]Centre of the Region Haná for Biotechnological and Agricultural Research, Faculty of Science, Palacký University and Institute of Experimental Botany ASCR, Olomouc, Czech Republic

**Abstract** Shoot branching is a primary contributor to plant architecture, evolving independently in flowering plant sporophytes and moss gametophytes. Mechanistic understanding of branching is largely limited to flowering plants such as *Arabidopsis*, which have a recent evolutionary origin. We show that in gametophytic shoots of *Physcomitrella*, lateral branches arise by re-specification of epidermal cells into branch initials. A simple model co-ordinating the activity of leafy shoot tips can account for branching patterns, and three known and ancient hormonal regulators of sporophytic branching interact to generate the branching pattern- auxin, cytokinin and strigolactone. The mode of auxin transport required in branch patterning is a key divergence point from known sporophytic pathways. Although PIN-mediated basipetal auxin transport regulates branching patterns in flowering plants, this is not so in *Physcomitrella*, where bi-directional transport is required to generate realistic branching patterns. Experiments with callose synthesis inhibitors suggest plasmodesmal connectivity as a potential mechanism for transport.

*For correspondence: cjh97@cam.ac.uk

†These authors contributed equally to this work

Competing interests: The authors declare that no competing interests exist.

## Introduction

The radiation of the vascular plants was underpinned by the innovation of a branching growth habit in the shoots of their last shared common ancestor around 430 million years ago (*Langdale and Harrison, 2008*; *Edwards et al., 2014*). The earliest vascular plants branched by bifurcation, involving the even partitioning of the growing tip into two new shoot tips (*Sussex and Kerk, 2001*; *Harrison et al., 2005*, *2007*; *Langdale and Harrison, 2008*; *Harrison and Langdale, 2010*; *Ligrone et al., 2012*; *Edwards et al., 2014*; *Tomescu et al., 2014*). Subsequent diversification within the seed plants was underpinned by the evolution of lateral (axillary) branching (*Sussex and Kerk, 2001*; *Langdale and Harrison, 2008*), in which buds initiate in leaf axils but may then become dormant until receiving environmental or internal cues to promote their activation and growth. Lateral branching thus permits finely tuned regulation of plant architecture and space filling in response to the environment (*Bergamini and Peintinger, 2002*; *Domagalska and Leyser, 2011*). Whilst branching by bifurcation is prevalent in non-seed vascular plant sporophytes and many bryophyte gametophytes, a capacity for lateral branching evolved by convergence in mosses and primed their diversification (*Farge-England, 1996*).

The mechanisms regulating lateral branching are well studied in flowering plant sporophytes, in which the initiation of axillary meristems is regulated by a drop in auxin levels and a rise in cytokinin

**eLife digest** Most land plants have shoots that form branches and plants can regulate when and where they grow these branches to best exploit their environment. Plants with flowers and the more ancient mosses both have branching shoots, but these two groups of plants evolved to grow in this way independently of each other. Most studies on shoot branching have focused on flowering plants and so it is less clear how branching works in mosses.

Three plant hormones—called auxin, cytokinin and strigolactone—control shoot branching in flowering plants. Auxin moves down the main shoot of the plant to prevent new branches from forming. This movement is controlled by the PIN proteins and several other families of proteins. On the other hand, cytokinin promotes the growth of new branches; and strigolactone can either promote or inhibit shoot branching depending on how the auxin is travelling around the plant.

Coudert, Palubicki et al. studied shoot branching in a species of moss called *Physcomitrella patens*. The experiments show that cells on the outer surface of the main shoot are essentially reprogrammed to become so-called 'branch initials', which will then develop into new branches. Next, Coudert, Palubicki et al. made a computational model that was able to simulate the pattern of shoot branching in the moss.

Further experiments supported the predictions made by the model. Coudert, Palubicki et al. found that, as in flowering plants, auxin from the tip of the main shoot suppresses branching in the moss, and cytokinin promotes branching. The experiments also showed that strigolactone inhibits shoot branching, but its role is restricted to the base of the shoots. The model predicts that, unlike in flowering plants, auxin must flow in both directions in moss shoots to produce the observed patterns of shoot branching. Also, the experiments suggest that the PIN proteins and another group of proteins that control the movement of auxin do not regulate shoot branching in moss. Instead, it appears that auxin may move through microscopic channels that link one moss cell to the next.

Coudert, Palubicki et al.'s findings suggest that both flowering plants and mosses have evolved to use the same three hormones to control shoot branching, but that these hormones interact in different ways. One key next step will be to find out how auxin is transported during shoot branching in moss by manipulating the opening of the channels between the cells. A further challenge will be to find out the precise details of how the hormones control the activity of the branch initial cells.

levels as leaves initiate from the apical meristem (*Wang et al., 2014a*, *2014b*). Classical decapitation experiments showed that the main shoot apex exerts an inhibitory effect over subsequent branch outgrowth, a phenomenon known as apical dominance. Replacing decapitated apices with lanolin impregnated with phytohormones showed that auxin can mediate this inhibition, and that its action can be antagonized by cytokinin application to buds (*Thimann and Skoog, 1933*; *Wickson and Thimann, 1958*; *Cline, 1991*). Strigolactone has recently been identified as a third hormonal regulator of branching, exerting an inhibitory or stimulatory effect on branch outgrowth depending on the auxin transport status of the plant (*Gomez-Roldan et al., 2008*; *Umehara et al., 2008*; *Shinohara et al., 2013*).

The mechanism by which the co-ordinated action of auxin, cytokinin and strigolactone regulates branching is not yet fully clear, but regulated auxin transport plays a key role (*Crawford et al., 2010*; *Domagalska and Leyser, 2011*). Auxin is synthesized in young expanding leaves and is transported basipetally by several transporters including PIN-FORMED1 (PIN1) polar auxin efflux carriers to generate the polar auxin transport stream in the stem (*Gälweiler et al., 1998*; *Ljung et al., 2001*). The suppressive action of auxin on lateral branch outgrowth is mediated indirectly (*Morris, 1997*; *Booker et al., 2003*), leading to the hypothesis that a second messenger (for example cytokinin) could act as an intermediary between auxin and bud activation (*Morris, 1997*; *Booker et al., 2003*). However, a dominant bud need not be apical, and more recent work suggests that buds compete to export auxin into the main auxin transport stream and that such competition is enhanced by a suppressive action of strigolactones on PIN1-mediated auxin transport (*Crawford et al., 2010*; *Prusinkiewicz et al., 2010*; *Shinohara et al., 2013*). Thus, PIN-mediated auxin transport is a key integration point in the regulation of branching patterns.

Despite the pivotal contribution of branching pattern innovations to the evolution of plant architecture, the mechanisms underlying the evolution of branching are poorly understood

(*Sussex and Kerk, 2001*). Auxin transport assays and decapitation experiments used in conjunction with pharmacological treatments in the lycophyte, *Selaginella*, suggest that auxin (acting via polar transport) and cytokinin are conserved regulators of branching in vascular plant sporophytes (*Williams, 1937*; *Wochok and Sussex, 1973*, *1975*; *Sanders and Langdale, 2013*). Although bryophyte sporophytes do not normally branch, there is a detectable basipetal auxin transport stream in mosses (*Fujita et al., 2008*). In *Physcomitrella*, disruption of auxin transport by application of polar transport inhibitors or perturbations in PIN function can induce a branching form (*Fujita et al., 2008*; *Bennett et al., 2014b*) that closely resembles branching forms in the early fossil record, and such branching forms have also been reported as rare natural liverwort variants (*Bower, 1935*). These data suggest that bulk basipetal polar auxin transport is a conserved regulator of land plant sporophyte development, and point to a potential contribution of PIN-mediated polar auxin transport to the evolution of sporophytic branching (*Fujita et al., 2008*; *Bennett et al., 2014b*).

The extent of conservation between sporophytic and gametophytic branching mechanisms is unknown. Classical decapitation experiments revealed apical dominance in mosses and showed that as in flowering plants, a suppressive role of the apex on lateral branching acts via auxin and can be antagonized by cytokinin (*von Maltzahn, 1959*). Whilst *Physcomitrella* PIN proteins are plasma-membrane targeted polar auxin transporters with many roles in gametophore development (*Bennett et al., 2014a*, *2014b*; *Viaene et al., 2014*), bulk basipetal auxin transport is not detected with radiolabelled auxin transport assays in moss gametophores (*Fujita et al., 2008*; *Fujita and Hasebe, 2009*), suggesting that auxin transport patterns are not shared between sporophytes and gametophytes. The contribution of cytokinins to gametophore branching has not been extended beyond the pharmacological approaches mentioned above (*von Maltzahn, 1959*), and strigolactone biosynthesis *ppccd8* mutants have increased branching in protonemal tissues but no reported gametophore defects (*Proust et al., 2011*). These data have led to the hypothesis that distinct developmental mechanisms have been recruited to regulate gametophyte and sporophyte shoot architecture in evolution (*Fujita et al., 2008*; *Fujita and Hasebe, 2009*).

Here, we investigated the hormonal regulation of lateral gametophore branching in the moss *Physcomitrella patens*. We present a simple model in which apical auxin sources interact via non-polar auxin transport with hormonally regulated global and local sensitivities to auxin elsewhere in the moss gametophore, accurately reproducing observed branching patterns. Our work suggests that three conserved hormonal cues have been recruited independently in evolution to produce a convergent branching morphology, but that their co-ordinated action is integrated by different mechanisms to those in flowering plants.

## Results

### Branch initials are specified de novo on the gametophore epidermis

To determine the manner of branch initiation in *Physcomitrella*, we first undertook a morphological and histological analysis. We found rhizoids initiating in all leaf axils beyond a certain distance from the main gametophore apex (*Figure 1A–D*) but branches were present in only a subset of axils (*Figure 1E–P*). In contrast to reports from other species that have identified stereotypical and taxon-specific branch initiation patterns (*Berthier et al., 1965*; *Berthier, 1970*), we found no regular spacing in the pattern of initiation of *Physcomitrella* branches. We were unable to detect any evidence of dormancy, suggesting that branch initiation and outgrowth are not distinct developmental processes (*Figure 1A–D*). At the earliest stage of branch initiation that we were able to detect, branches were manifest as a single apical cell surrounded by leaf initials and adjacent to rhizoids (*Figure 1E–H*), appearing to differentiate from the epidermis (*Figure 1G–H*). At later stages of development, more rhizoids developed at the base of each branch and newly formed leaves were distinguishable (*Figure 1I–L*). Well developed lateral branches were morphologically similar to the main gametophore axis (*Figure 1M*), and transverse sections cut at the base of lateral branches showed that each branch persisted as a superficial projection; there was no continuity between the conducting tissue of the lateral branches and the conducting tissue of the main stem (*Figure 1N–P*). We reasoned that the largest branches would have initiated first and evaluated the sequence of branch initiation by dissecting branches from 6 week old gametophores, and ranking their size relative to their position (*Figure 1Q*). These morphological data suggest that in *Physcomitrella* branches initiate de novo from the epidermis of the gametophore axis and that initiation is usually, but not invariably, acropetal.

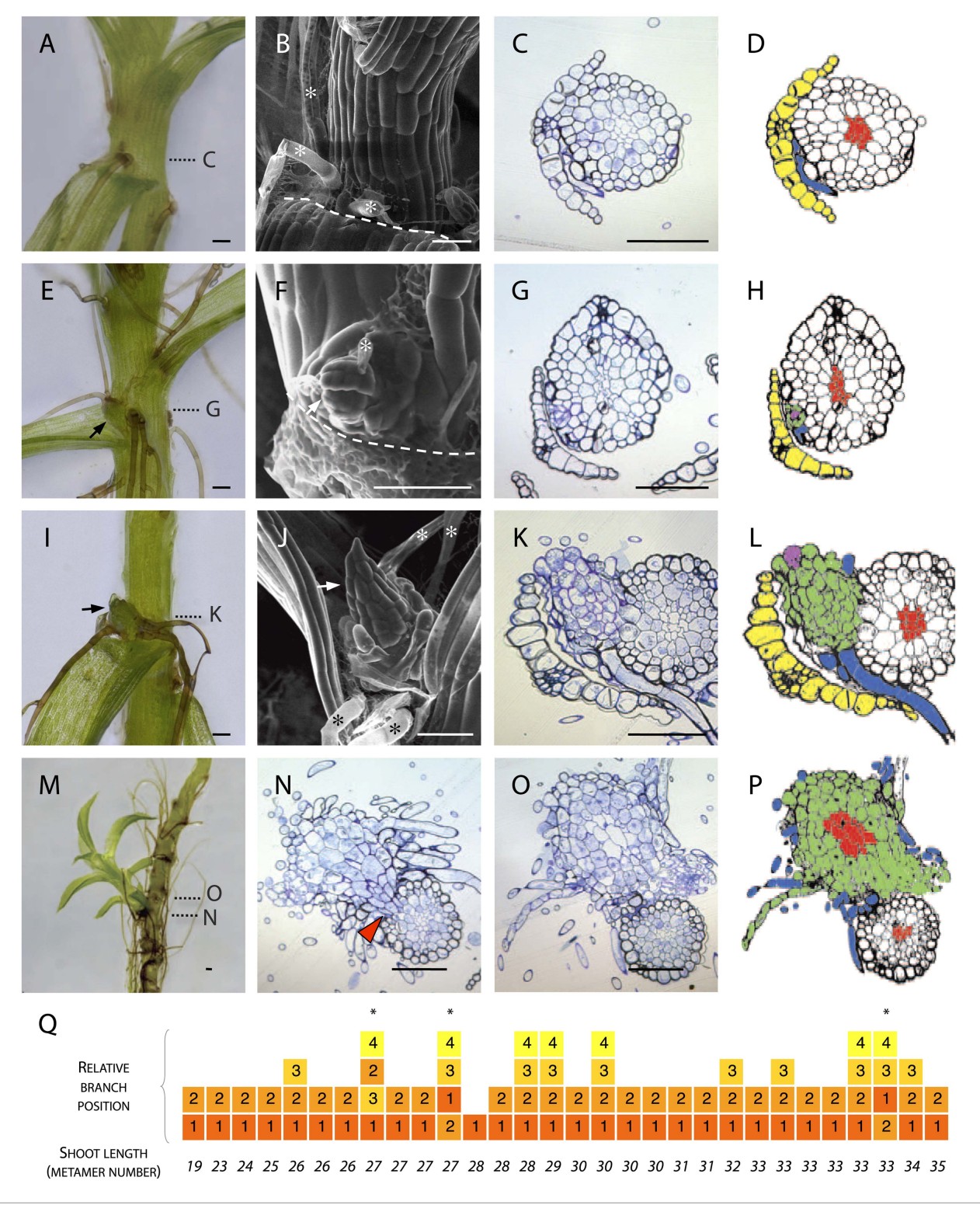

**Figure 1**. Branches initiate from the epidermis or outermost cortical cell layer in *Physcomitrella*. (A–D) Although all leaf axils contained axillary rhizoids, branches were absent in most. (A) Light micrograph, (B) scanning electron microscope (SEM) micrograph, (C) transverse histological section and (D) corresponding line drawing showing a rhizoid (blue) in the axil of a leaf (yellow). Gametophore conducting tissues are shown in red. The label **C** in (A) indicates the approximate plane of section in (C). (E–H) At the earliest detectable stage of branching, each branch comprised a single apical cell surrounded by leaf initials and was adjacent to one or several developing rhizoids. (E) Light micrograph, (F) SEM micrograph, (G) transverse histological

*Figure 1. continued on next page*

*Figure 1. Continued*

section and (**H**) corresponding line drawing showing a branch apical cell (pink) surrounded by leaf initials (green) and adjacent to an initiating rhizoid (blue). The label **G** in (**E**) indicates the approximate plane of section in (**G**). (**I–L**) At a later stage, growing buds had well-developed leaves. (**I**) Light micrograph, (**J**) SEM micrograph, (**K**) transverse histological section and (**L**) corresponding line drawing showing a well-developed bud (green) and its apical cell (pink) in the axil of the leaf (yellow). The label **K** in (**I**) indicates the approximate plane of section in (**K**). (**M–P**) Well-developed branches persisted as superficial projections, there was no continuity between the conducting tissue system of the branch and the gametophore axis. (**M**) Light micrograph of a lateral branch on a gametophore whose leaves have been removed. (**N**) Transverse histological section at the junction point of the lateral branch with the main gametophore axis (indicated by **N** in [**M**]), the red arrowhead shows cortical tissue and the absence of continuity between conducting tissue systems. (**O**) Transverse histological section and (**P**) corresponding line drawing above the junction point (indicated by **O** in [**M**]) where the conducting tissues (red) of the lateral branch (green) and the main axis (white) can be seen. (**Q**) The sequence of branch initiation in thirty 6 week old gametophores. For all images, arrows show lateral buds, dotted lines indicate the level of corresponding histological sections, asterisks mark rhizoids, dashed lines mark the boundary between the stem and the detached leaf. Scale bars = 100 µm.

## Branch initiation is patterned

To determine how branches are distributed in *Physcomitrella*, the leaves were removed from 60 wild-type (WT) gametophores grown on sterile BCD + AT medium for 5 or 7 weeks (*Figure 2A*). The position of lateral branches was recorded (*Figure 2B*), showing an uneven distribution (*Figure 2C*). The formation of a minimum of 18 leaves was required prior to branch initiation, and an apical portion devoid of branches was maintained at a similar length throughout development. We termed the barren apical portion of the gametophore the apical inhibition zone (AIZ), and the portion of the gametophore in which branches initiated was termed the branching zone (BZ). Reasoning that each of these aspects of the branch distribution might be under regulatory control, we sought to determine how the branch initiation pattern deviated from a random distribution. We simulated a dataset of 60 gametophores in which branching occurs stochastically with a probability of 5% to obtain a similar number of branches to WT (*Figure 2D*, 'Materials and methods'). We defined a gametophore as a 1D series of metamers where each metamer consists of a section of the main gametophore axis and a leaf, and growth proceeds at a fixed rate adding new metamers as the simulation progresses. Although it was immediately apparent that the simulated dataset lacked apical inhibition, potential differences in branch spacing in the branching zone between WT and simulated datasets were not obvious. We therefore calculated and compared the mean minimum distance between branches in the branching zone of WT (4.48 ± 1.48) and simulated stochastically branching shoots (3.62 ± 0.41). In the WT dataset, branches were more evenly dispersed than expected at random (p-value < 0.05; *Figure 2D*), supporting the notion that the distribution of branches within the BZ is regulated. Even if an apical inhibition zone is introduced into a random model of branching, for example by assuming a branching competency that requires a minimum metamer age, it cannot produce a realistic branching pattern (*Figure 2E*, *Figure 2—figure supplement 1*).

## A model integrating the effect of a notional mobile apical cue with sensitivity to that cue elsewhere in the gametophore can account for branching patterns

Previous studies in mosses have shown that the main gametophore apex exerts an inhibitory effect over branching that acts via auxin (*von Maltzahn, 1959*; *Nyman and Cutter, 1981*). To help us understand branch patterning, we generated a regulatory model of moss branching which assumes the main apex and newly forming branch apices are sources of auxin (assumption 1; *Figure 2F*). The terminal apex is represented by a red square and produces new metamers represented by grey squares (*Figure 2G*; rule 1). The bottom metamer is always defined as the first of the 1D metamer file representing the gametophore in the model. The concentration of auxin in a metamer is expressed with the parameter $c$, and auxin levels in the terminal apex and lateral apices are set to fixed values of $H_{apex}$ and $H$ respectively; thus auxin is produced to a constant concentration. Biologically $H_{apex}$ and $H$ represent a combination of inputs to regulate auxin levels including auxin synthesis, transport and decay, and any potential feedback between these processes.

We assume that auxin moves from sites of production into neighboring metamers, changing the auxin concentration in those metamers (assumption 2; *Figure 2H*). Such changes per metamer are expressed in the model with the following equation:

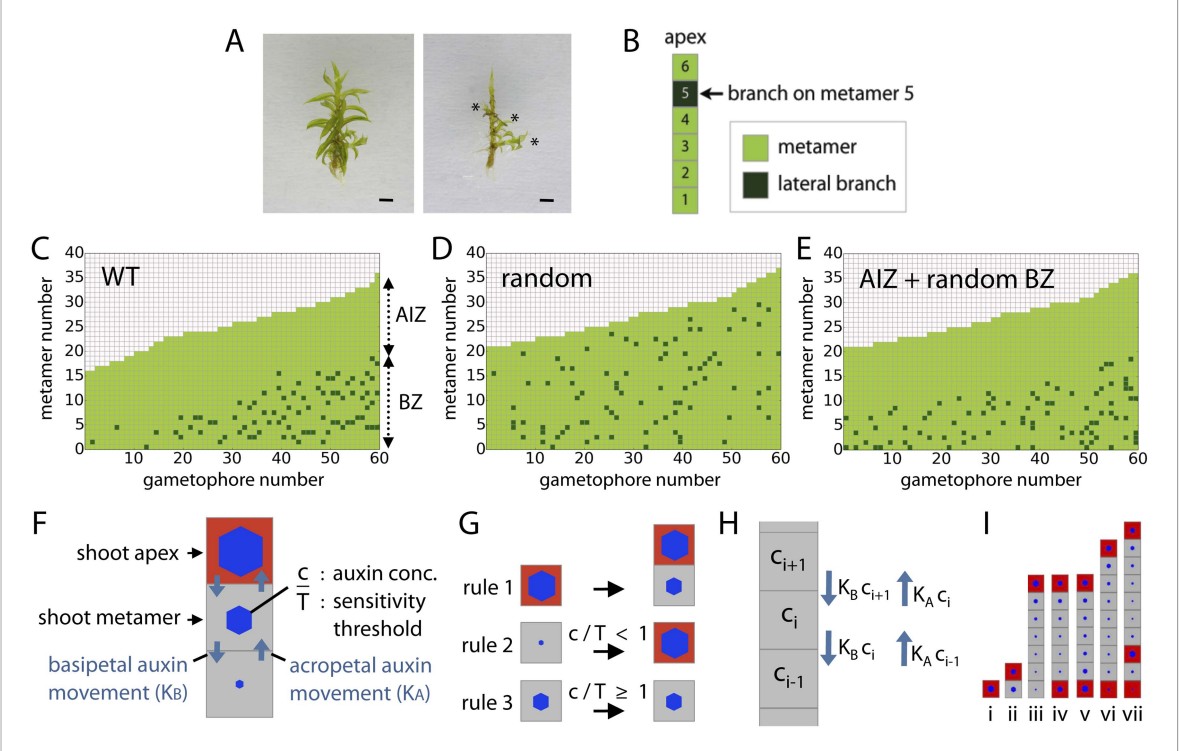

**Figure 2.** Branching patterns are non-random. (**A**) A wild-type gametophore before (left) and after (right) removing the leaves. Asterisks indicate lateral branches. Scale bar = 1 mm. (**B**) Each gametophore is represented as 1-D series of metamers (light green squares) and lateral branch position is indicated in dark green. (**C–E**) Branching patterns of 60 gametophores ordered by increasing size showing the apical inhibition zone (AIZ) and branching zone (BZ). Data from wild-type (**C**), stochastic simulation (**D**) and stochastic simulation with imposed apical inhibition (**E**) had similar branch numbers but different branch distributions. (**F–I**) Assumptions and replacement rules of the computational model. (**F**) Each gametophore starts off as an apex (red), which produces auxin and grows by producing metamers (grey) containing auxin at concentration $c$. (**G**) illustrates replacement rules used in simulations. 1: an apex (red) regularly produces metamers (grey). 2: if auxin concentration $c$ in a metamer falls below the auxin sensitivity threshold $T$, then the metamer becomes a branch and an auxin source (red). 3: if auxin concentration $c$ is higher than $T$, then branch formation is inhibited. (**H**) The auxin concentration $c$ within a metamer reflects acropetal transport defined by the constant $K_A$ and basipetal transport defined by the constant $K_B$. (**I**) Stages of growth in a single simulated gametophore showing that, as new metamers are added, the ratio of $c/T$ drops towards the base, allowing a branch to initiate (compare iii to iv). The new apex becomes an auxin source and can export auxin both up and down the gametophore (v). This results in the formation of an auxin minimum further up the gametophore axis and a second branch initiates (vi to vii).

The following figure supplements are available for figure 2:

**Figure supplement 1.** Comparison of the WT branching pattern plot with different model outputs.

**Figure supplement 2.** Stages of growth in a single simulated gametophore showing that as new metamers are added, the ratio of $c/T$ drops towards the base, allowing a branch to initiate (compare v to vi).

$$\frac{dc_i}{dt} = K_B(c_{i+1} - c_i) + K_A(c_{i-1} - c_i) - vc_i, \quad (1)$$

where $c$ is the concentration of auxin, $t$ is the time interval between simulation steps, $K$ represents an auxin transport constant, $i$ represents metamer indexing and $v$ represents auxin decay.

Since metamer indexing increases from basal to apical metamers, parameter $K_B$ in equation (**Langdale and Harrison, 2008**) always represents basipetal movement of auxin and parameter $K_A$ represents acropetal movement of auxin. If $K_A = K_B$, auxin movement between metamers is not biased acropetally or basipetally, and equation (**Langdale and Harrison, 2008**) conforms to Fick's second law of diffusion. If $K_A \neq K_B$, auxin movement is directionally biased.

A lateral branch can only initiate if the concentration of auxin in a metamer falls below a threshold $T$ (assumption 3), or $c/T < 1$ (**Figure 2G**; rules 2 and 3). Points of branch initiation and insertion are

abstracted as red squares in the metamer file produced by the main apex (*Figure 2I*). Although branch outgrowth is not depicted, it is also simulated as illustrated in *Figure 2—figure supplement 2*. The ratio *c*/*T* is represented by blue hexagons (*Figure 2F,G,I*).

Parameters $H_{apex}$, *H* and *T* are determined for each gametophore in a series before simulations start and are stochastically attributed values within a specified range. Parameter values may therefore differ between, but not within gametophores.

The model incorporates a constant notional level of auxin decay represented by parameter *v* (see 'Materials and methods'). Therefore metamers that are further away from the apex are older, and have less auxin derived from the terminal apex than newer metamers, and a smaller value of *c*/*T* (*Figure 2I*, *Figure 2—figure supplement 2*). This accounts for branch initiation towards the gametophore base. The new lateral branch apices produce auxin, which moves to metamers adjacent to the branch insertion point (*Figure 2I$_v$*). As the gametophore grows, a local auxin minimum appears between active apices (*Figure 2I$_{vi}$*) and another branch initiates (*Figure 2I$_{vii}$*).

## Bidirectional transport of an apical cue such as auxin is required to generate a realistic branching pattern

To determine whether the model can capture WT branching patterns, a series of 60 gametophores with 20–40 leaves was simulated. In line with published data showing bulk basipetal auxin transport to be undetectable in moss gametophores, parameter exploration showed that the WT branch distribution was best approximated by ratios of $K_A/K_B = 1$ (*Table 1*). A small bias in the direction of transport had a major impact on the distribution of branches *Figure 3*). Where $K_A/K_B > 1$, branches initiated both acropetally and basipetally and apical inhibition was lost (*Figure 3A,B*, *Videos 1, 2*). Where $K_A/K_B < 1$, the branch density in the basal portion of each gametophore and the size of the apical inhibition zone were both significantly increased (*Figure 3C,D*, *Videos 3, 4*). When $K_A/K_B$ was set to reflect levels of basipetal transport detected in *Arabidopsis* ($K_A/K_B = 1/100$), basal branches activated consecutively in the branching zone (*Figure 3D*). Thus bidirectional transport was required in order to generate an apical inhibition zone comparable in metamer number to those in real plants, and to generate a branching zone with a branch distribution similar to the distribution in real plants (compare *Figures 2C* to *Figure 3E,F*, and see *Figure 2—figure supplement 1C* for quantitative comparison. See also *Video 5*).

A point of convergence between the model and branch distribution data from real plants was that in both, branch initiation in adjacent metamers was occasionally observed (compare *Figure 3F* to *Figures 4B,C,D*, *5A*, 8A). We note that the frequency of such initiation is lower in data from real plants (8/210 gametophores using combined data from *Figures 2C, 4B, 5A*, 8A) than in the model (7/60 gametophores with data shown in *Figure 3F*), and it is possible that there are mechanisms to prevent adjacent initiations in plants. A further point of comparison relates to the sequence of branch activation (*Figure 1Q*), which we originally assumed to be acropetal. Although branches normally initiate in an acropetal series, the model predicted that minima of *c*/*T* should also form out of series (see *Figure 2—figure supplement 2*), and we also found that branches can initiate out of series in real plants (*Figure 1Q*).

A point of divergence between branch patterns predicted by the model and data from real plants was that given bidirectional auxin transport, model simulations constitutively activated branches in the basal

**Table 1**. Parameter values for the models shown in *Figures 3, 6, 7*

|  | Model wild-type | Model *shi2-1* | Model *SHI ox-5* | Model *CKX2oe* | Model *IPT1oe* |
|---|---|---|---|---|---|
|  | (*Figure 3F*) | (*Figure 6A*) | (*Figure 6B*) | (*Figure 7A*) | (*Figure 7B*) |
| $H_{apex}$ | $\mu = 80$, $\sigma = 20$ | $\mu = 48$, $\sigma = 12$ | $\mu = 240$, $\sigma = 60$ |  |  |
| *H* | $\mu = 20$, $\sigma = 4.5$ | $\mu = 12$, $\sigma = 2.7$ | $\mu = 60$, $\sigma = 13.5$ |  |  |
| *T* | $\mu = 3.0$, $\sigma = 0.8$ |  |  | $\mu = 0.45$, $\sigma = 0.2$ | $\mu = 5.4$, $\sigma = 0.8$ |
| *v* | 0.01 |  |  |  |  |
| $K_A$ | 0.05 |  |  |  |  |
| $K_B$ | 0.05 |  |  |  |  |

Parameter $\mu$ represents the mean and $\sigma$ the variance of the normally distributed stochastic variables *H* and *T*. Parameter values left blank are identical to the wild-type model.

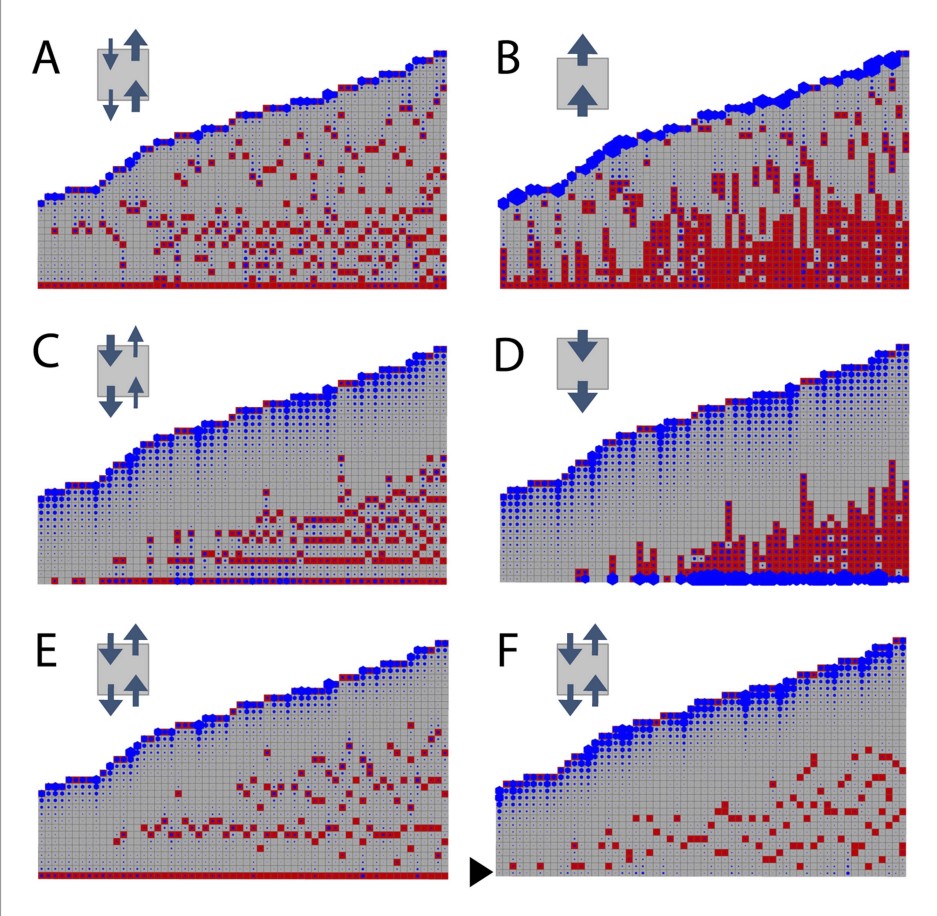

**Figure 3**. Simulated branching patterns are sensitive to changes in the direction of auxin transport and predict a basal branching inhibitor. (**A**–**B**) Output of simulations with mainly acropetal auxin transport, set to $K_A/K_B = 3$ (**A**) or $K_A/K_B = 100$ (**B**). (**C**–**D**) Output of simulations with predominantly basipetal transport, set to $K_A/K_B = 1/3$ (**C**) or $K_A/K_B = 1/100$ (**D**). (**E**, **F**) Output of simulations with bi-directional auxin transport set to $K_A/K_B = 1$. Without local basal reduction of the branching threshold ($T$, black arrow), the most basal metamer always produced a branch (**E**). A reduction of the branching threshold in the basal metamers was required to generate a realistic WT branching pattern (**F**). For all insets, arrow sizes indicate the relative amount of basipetal and acropetal auxin transport. Red indicates gametophore and branch apices, and blue indicates the auxin concentration ($c$) relative to $T$.

metamer (*Figures 2I*, *3E*). A phenomenon that was not observed in real plants (compare *Figure 2C* to *Figure 3E*). This led us to hypothesize that branching could be locally suppressed at the gametophore base by an unknown inhibitory signal. We adjusted the model by locally reducing the branching threshold in the most basal metamers of the gametophore (assumption 4; *Figure 2—figure supplement 2*). As a decrease in the value of $T$ drives up the ratio $c/T$, basal metamers less frequently reached conditions where $c < T$ than in simulations without the basal inhibitor. The frequency of branch initiation in the bottom five metamers of subsequent simulations was similar to the frequency of branch initiation in those metamers in real plants (*Figure 3F*, *Figure 2—figure supplement 1C* and *Video 6*).

Therefore, the model attained branch distribution patterns that were comparable to data from real plants (*Figure 2—figure supplement 1*), and generated testable hypotheses relating to the regulation of branch initiation.

## The main gametophore apex inhibits branching, acting via auxin

To test the assumptions of our model we undertook a series of experiments in *Physcomitrella*. We first sought to identify potential contributions of the main gametophore apex and auxin to the branching pattern by performing surgical experiments in which thirty gametophores were teased out of several

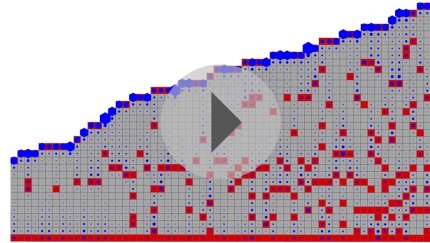

**Video 1.** (corresponds to *Figure 3A*) Simulation of branching activation pattern with mainly acropetal auxin transport (set to $K_A/K_B = 3$) and no basal branching inhibitor.

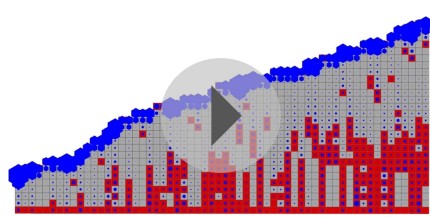

**Video 2.** (corresponds to *Figure 3B*) Simulation of branching activation pattern with mainly acropetal auxin transport (set to $K_A/K_B = 100$) and no basal branching inhibitor.

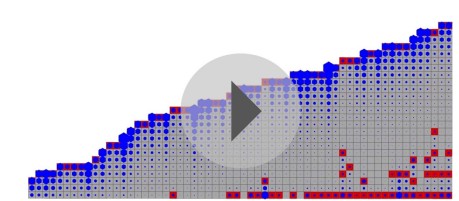

**Video 3.** (corresponds to *Figure 3C*) Simulation of branching activation pattern with mainly basipetal auxin transport (set to $K_A/K_B = 1/3$) and no basal branching inhibitor.

colonies and decapitated by removal of the top six metamers (the top third of the apical inhibition zone). The excised apices were replaced with lanolin or lanolin impregnated with indole-3-acetic acid (IAA) (*Figure 4A*). Thirty further gametophores were isolated and left intact as a control, and all sets were left to grow for a week. Whilst control gametophores continued to grow as normal (*Figure 4B*), the gametophores which had their apices replaced by lanolin discontinued growth of the main axis and initiated branches in the portion of the stem originally comprising the apical inhibition zone beneath the cut site (*Figure 4C,E*). In contrast, the gametophores that had their apices replaced by IAA impregnated lanolin had impaired branch initiation and development (*Figure 4D, E*). Whilst a total of around 70 branches initiated from lanolin treated control gametophores, around 50 branches initiated from IAA treated gametophores, and of these less than 10 showed normal development. The remainder were shorter than normal and were unable to initiate leaves (*Figure 4E*), a defect that we have previously observed in the main apex in gametophores treated with high concentrations of auxin (*Bennett et al., 2014b*). No difference in the spacing of branches in control or experimental treatments was detected (*Figure 4F*).

These results suggest the hypothesis that the shoot apex suppresses branch initiation by acting as an auxin source, which contradicts the low auxin activity levels in gametophore tips reported by *GH3::GUS* signal accumulation. However, the *GH3::GUS* reporter reflects downstream transcriptional outputs of auxin signalling, and may not accurately reflect the auxin distribution (*Brunoud et al., 2012*). We therefore quantified IAA levels along the gametophore axis by sampling the top six metamers, the apical inhibition zone minus the tip and the branching zone with the branches removed (*Figure 4A*). Despite some variability between replicates, we found that mean IAA levels were highest in the tip. We also undertook decapitation experiments in the *GH3::GUS* reporter line and found that the signal intensity dropped substantially in decapitated plants, consistent with the notion that the apex can affect auxin levels elsewhere in the gametophore. In combination, data shown in *Figure 4* support assumption one of our model by (i) showing that the main apex can regulate the branching pattern, (ii) providing evidence that the main apex is an auxin source, and (iii) showing that apically applied auxin can suppress branching.

## Auxin suppresses branching

To determine whether the suppression of branching by auxin is position dependent or whether auxin can act as a global suppressor of branching, 5 week old gametophores were immersed in an auxin solution for 24 hr and grown for a further 2 weeks prior to analysis of branching patterns (*Figure 5*). A

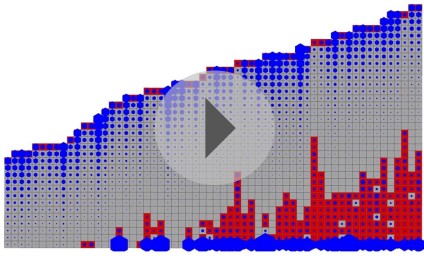

**Video 4.** (corresponds to *Figure 3D*) Simulation of branching activation pattern with mainly basipetal auxin transport (set to $K_A/K_B = 1/100$) and no basal branching inhibitor.

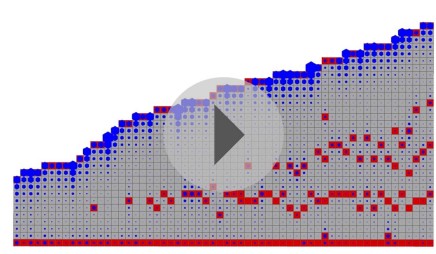

**Video 5.** (corresponds to *Figure 3E*) Simulation of branching activation pattern with equal basipetal and acropetal auxin transport (set to $K_A/K_B = 1$) and no basal branching inhibitor.

natural auxin, IAA, and two synthetic auxins with different metabolic and transport properties, 1-Naphthaleneacetic acid (NAA) and 2,4-Dichlorophenoxyacetic acid (2,4-D), were tested. For all the treatments, the size of the apical inhibition zone significantly increased, the branch number was strongly reduced and branch initiation in the branching zone significantly decreased, suggesting that auxin can act as a global suppressor of branching.

## Auxin biosynthesis mutants capture predicted effects of changing model values $H_{apex}$ and $H$

Although our model assumes that both the main and lateral apices can act as auxin sources ($H_{apex}$ and $H$ respectively), we were unable to isolate the effect of lateral apices in surgical experiments due to technical constraints. We therefore hypothesized that the effects of altering global levels of auxin synthesis could capture aspects of variance in both the primary and lateral gametophore tips (*Figure 6*, *Figure 6—figure supplement 1*). To test this hypothesis, we grew *Physcomitrella* mutants or transgenics in which auxin synthesis is diminished or increased (*Eklund et al., 2010*), and quantified their branching patterns with respect to model predictions and WT controls. A global decrease in auxin levels was simulated by reducing the values of $H_{apex}$ and $H$, and the predicted outcome was a reduction of the apical inhibition zone size, accompanied by an increase in branch density within the branching zone (*Figure 6A*). The branching phenotype of *Ppshi2-1*, an auxin-deficient mutant in which auxin synthesis at the gametophore apex is reduced (*Eklund et al., 2010*), matched this predicted outcome (*Figure 6C,E,G,I,J*, *Figure 6—figure supplement 1*, *Video 7*). Conversely, a global increase in auxin levels was simulated by increasing the values of $H_{apex}$ and $H$, generating the predictions that the apical inhibition zone size should increase and the branch density should decrease (*Figure 6B*). The branching pattern of *PpSHI ox-5*, a transgenic line with elevated auxin levels (*Eklund et al., 2010*), again matched the model output (*Figure 6D,F,H,I,J*, *Figure 6—figure supplement 1*, *Video 8*). Thus, by generating a decreased inhibition zone with more branches in the branching zone or an increased inhibition zone with fewer branches in the branching zone respectively, the effects of depleting or elevating *SHI*-mediated auxin biosynthesis captures predicted effects of changing the target auxin concentrations $H_{apex}$ and $H$ in the model.

## Cytokinin biosynthesis mutants capture predicted effects of changing model values $T$

A second assumption of our regulatory model is that the sensitivity to auxin (parameter $T$) modulates the branching pattern. In bud treatment experiments in flowering plants and moss, cytokinin antagonizes the suppressive effects of auxin in promoting branching (*Thimann and Skoog, 1933*; *Wickson and Thimann, 1958*; *von Maltzahn, 1959*), and we postulated that cytokinin could modulate auxin sensitivity with levels approximating to values of $T$ (*Figure 7*, *Figure 6—figure supplement 1*). Therefore, decreasing the values of $T$ in our model should have the converse effect to an increase in parameter values of $H$, and this effect should be observed in moss shoots by global reduction of cytokinin levels. To test this hypothesis, we grew *Physcomitrella* transgenics in which cytokinin degradation is increased, thereby decreasing cytokinin levels (*von Schwartzenberg et al., 2007*), and quantified their branching patterns (*Figure 7*). The predicted outcome of the model was an increase of the apical inhibition zone size and a reduction in

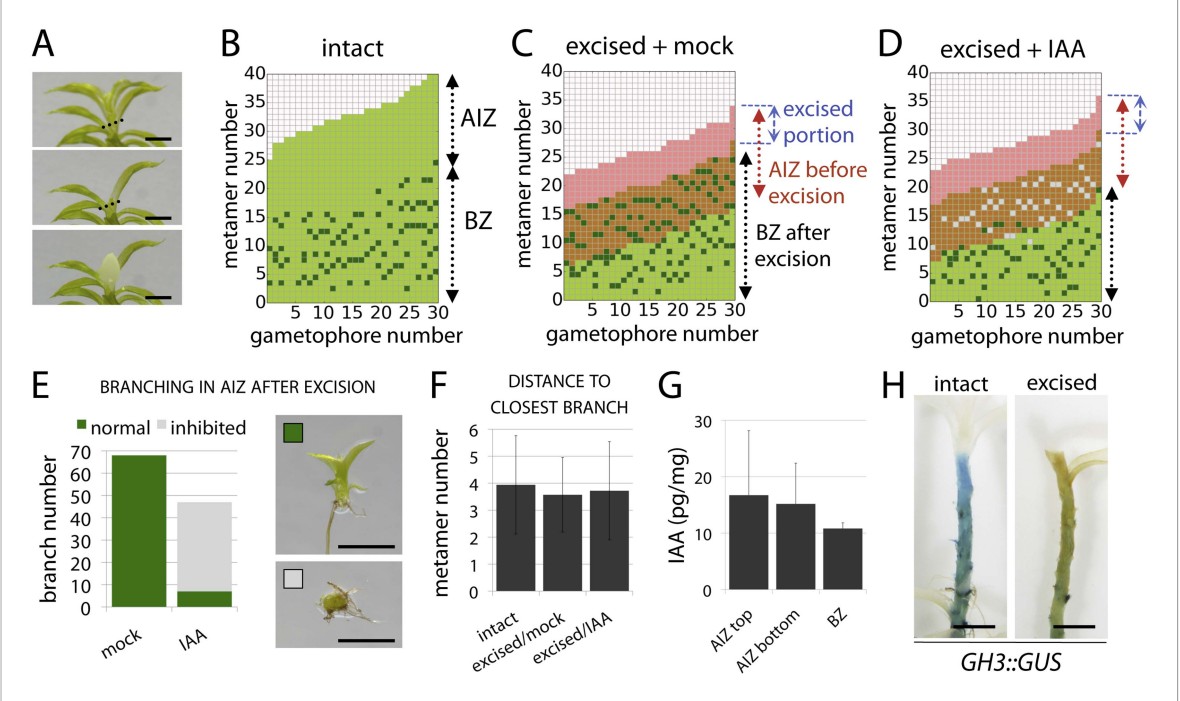

**Figure 4**. The main *Physcomitrella* gametophore apex is an auxin source that suppresses branching. (**A**) Gametophores were isolated from wild-type colonies (top), and the six top metamers were excised at the dotted line (middle) and replaced with lanolin or lanolin plus 1 mM auxin (bottom). Scale bar = 1 mm. (**B**) Branching pattern of 30 intact gametophores ordered by increasing size. AIZ, apical inhibition zone. BZ, branching zone. (**C**) Branching pattern of 30 gametophores 5 days after excision. Lateral branches activated in the portion of gametophore corresponding to the apical inhibition zone before excision, coloured in red. (**D**) Branching pattern of 30 gametophores 5 days after excision and replacement of the apex by a source of auxin. (**E**) The number of normal and defective branches formed in the apical inhibition zone of mock and IAA-treated gametophores. (**F**) Mean minimum metamer number between lateral branches was not affected by gametophore apex excision. (**G**) shows mean auxin levels quantified from five biological replicates; levels were are highest at the tip of the gametophore and decreased toward the base. (**H**) *GH3::GUS* expression was high in the apical inhibition zone (left) and was strongly reduced after decapitation (right). Scale bar = 1 mm.

branch number (*Figure 7A*). The branching pattern of a transgenic line that constitutively expresses the *Arabidopsis* cytokinin degradation gene, *CYTOKININ OXIDASE-2* (*CKX2oe*) (*von Schwartzenberg et al., 2007*), was quantified and found to be similar to the model output in having fewer branches and a larger apical inhibition zone (*Figure 7C,E,G,I,J*, *Figure 6—figure supplement 1*, *Video 9*). Conversely, increasing the values of *T* in the model should have similar effects to a global increase in cytokinin levels. To test this hypothesis, we generated *Physcomitrella ISOPENTENYL TRANSFERASE-1* (*von Schwartzenberg et al., 2007*) transgenics (*IPT1oe*) in which cytokinin levels are upregulated (*Figure 7—figure supplement 1*) and quantified their branching patterns. The predicted outcome of the model was a reduction of the apical inhibition zone size and an increase in the branch number (*Figure 7B*, *Video 10*), and the branching pattern in the *IPT1oe* line was similar (*Figure 7D,F,H,I,J*, *Figure 6—figure supplement 1*). Moreover, cytokinin application to gametophores was sufficient to promote meristematic cell formation and proliferation along the gametophore axis (*Figure 7K–N*). Thus cytokinin is necessary for and promotes branching and lateral meristem activity.

## Neither PINs nor ABCB/PGPs contribute strongly to branching patterns

A third assumption of our regulatory model is that auxin moves between metamers, and the output of the model was found to be very sensitive to changes in the direction of auxin transport. Realistic branching patterns were only generated when there was relatively even basipetal and acropetal transport within simulated gametophore axes (*Figure 3*), a result that is consistent with the absence of detectable bulk basipetal transport in *Physcomitrella* gametophores (*Fujita et al., 2008*). The molecular functions of canonical PIN proteins are conserved between flowering plants and mosses (*Bennett et al., 2014b*; *Viaene et al., 2014*), and *Physcomitrella* PINs have been demonstrated to

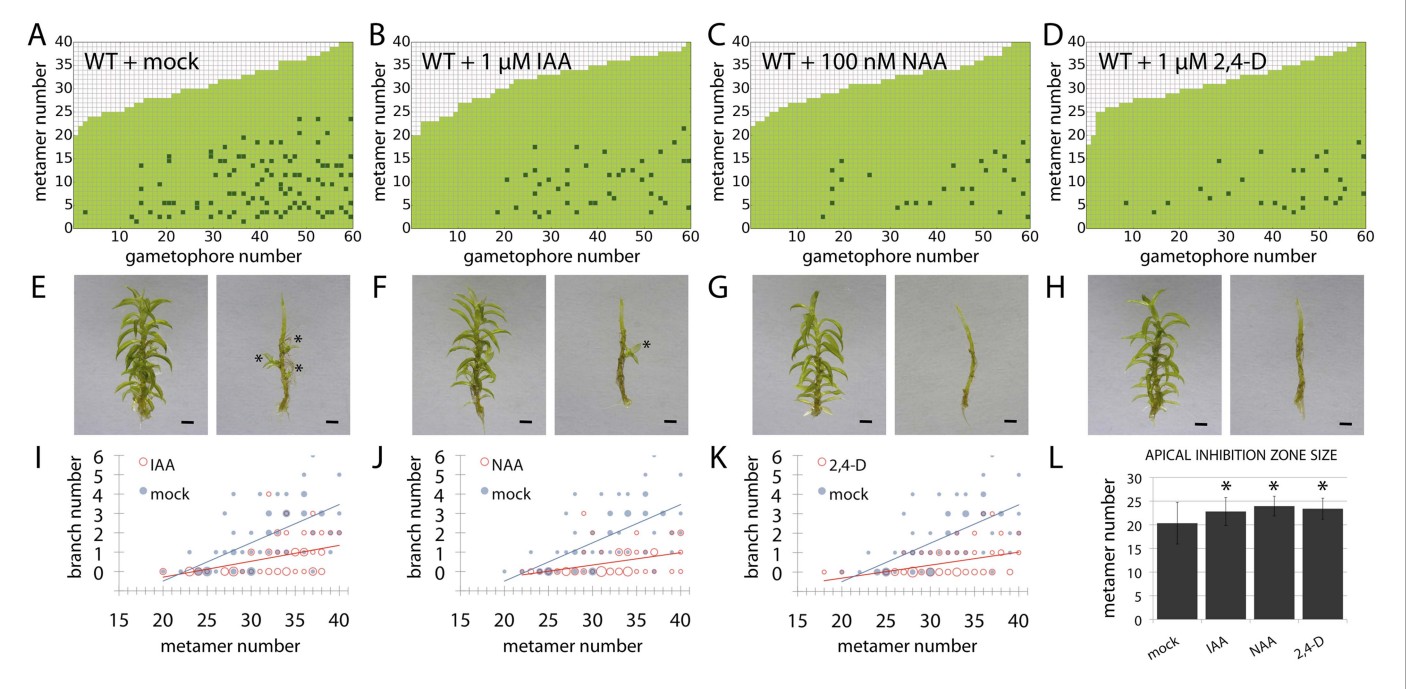

**Figure 5**. Globally applied exogenous auxins inhibit branch initiation. (**A–D**) Branching patterns in wild-type gametophores grown for 5 weeks, immersed for 24 hr in mock (**A**), 1 μM IAA (**B**), 100 nM NAA (**C**) or 1 μM 2,4-D (**D**) and grown for another 2 weeks. (**E–H**) mock (**E**), IAA (**F**), NAA (**G**) and 2,4-D (**H**) treated gametophores before (left) and after (right) removing the leaves, with asterisks indicating lateral branches. Scale bar = 1 mm. (**I–K**) Bubble plots showed that branch number decreased in response to IAA (**I**), NAA (**J**) and 2,4-D (**K**) compared with mock-treated gametophores. Gametophore length is depicted as the number of metamers and the bubble area is proportional to the number of gametophores with a similar branch number at a particular length. Ordinary least squares regression was used to test whether the relationship between branch number and gametophore leaf number depended on treatment (see 'Material and methods'), and for (**I**) the best fitting model was $B = (-4.45 + 2.52X) + (0.2 - 0.12X)L$ meaning that IAA treatment significantly differed from mock treatment ($p < 0.01$). For (**J**) the best fitting model was $B = (-4.45 + 2.87X) + (0.2 - 0.13X)L$; NAA treatment significantly differed from mock treatment ($p < 0.001$). For (**K**) the best fitting model was $B = (-4.45 + 2.8X) + (0.2 - 0.13X)L$; 2,4-D treatment significantly differed from mock treatment ($p < 0.001$). (**L**) The apical inhibition zone size increased in response to IAA, NAA and 2,4-D (mean ± SD; bilateral t-test different from mock control, *$p < 0.05$).

localise both proximally and distally within leaf cells (*Viaene et al., 2014*). PINs therefore provide one potential mechanism to account for bi-directional auxin transport in the regulation of branching. We previously engineered *Physcomitrella pin* mutants in a *GH3::GUS* background (*Bennett et al., 2014b*), and found that whilst branching patterns in the *GH3::GUS* line were similar to WT controls (*Figure 8A*), *pin* mutants had mildly disrupted branching patterns (*Figure 8A–H*). The apical inhibition zone was shorter in *pina pinb* mutants than in *GH3::GUS* controls (*Figure 8I*). Branch number was slightly increased in *pinb* and *pina pinb* double mutants (*Figure 8J–L*) and the mean minimum distance between branches was reduced in *pina*, *pinb* and *pina pinb* double mutants (*Figure 8M*). To rule out the possibility that residual PIN-mediated auxin transport could be responsible for the mild effects of *pin* mutants on branching patterns, mutants were grown on medium supplemented with 5 μM NPA (*Figure 8N–W*). In both *GH3::GUS* and *pina pinb* lines, branching patterns were found to be similar in mock and NPA-treated plants. Thus PIN-mediated auxin transport is not a major contributor to branching patterns.

A second class of auxin efflux transporters controlling development which interact with PIN proteins to regulate auxin distributions are the ATP-binding cassette protein subfamily B (ABCB/PGP) transporters (*Cho and Cho, 2013*). ABCB/PGP proteins have non-polar plasma membrane localizations and could therefore putatively generate the bi-directional auxin transport required by the model to regulate branching. 10 genes have been identified in the *Physcomitrella* genome (*Carraro et al., 2012*). To test the hypothesis that ABCB/PGPs regulate *Physcomitrella* branching, we analysed branching patterns in WT gametophores grown on an ABCB/PGP inhibitor (*Kim et al., 2010*), 2-[4-(diethylamino)-2-hydroxybenzoyl]benzoic acid (BUM) at a 5 μM concentration

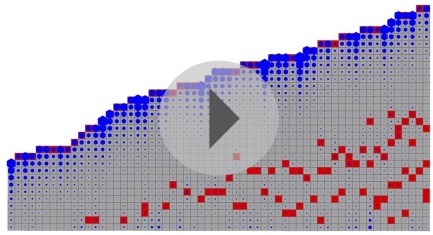

**Video 6.** (corresponds to *Figure 3F*) Simulation of branching activation pattern with equal basipetal and acropetal auxin transport (set to $K_A/K_B = 1$) and with basal branching inhibitor.

(*Figure 8X–AB*). We found no difference between mock and BUM-treated plants suggesting that ABCB/PGP proteins do not regulate branching in *Physcomitrella*.

## Callose synthesis inhibitors affect branching

Although the contributions of plasma membrane-targeted PIN and ABCB/PGP auxin transporters to plant development are well accepted (*Petrásek and Friml, 2009*), recent work in *Arabidopsis* has suggested that symplastic auxin transfer via plasmodesmata may also contribute to development by maintaining auxin concentration gradients between cells (*Han et al., 2014*). Symplastic permeability is under active control via callose deposition such that increased deposition blocks plasmodesmatal connections, and decreased deposition opens them up to allow greater permeability and higher rates of diffusion (*Han et al., 2014*). As the apolar auxin transport required to generate realistic branching patterns in our model is equivalent in principle to diffusion, we hypothesized that a callose-dependent mechanism could regulate branching. To test this hypothesis, we grew *Physcomitrella* on a callose biosynthesis inhibitor, 2-deoxy-glucose (DDG), and found that its application did not cause general defects at the concentrations used. We therefore compared the branching patterns to model predictions arising from a simulated increase in the rate of auxin movement (*Figure 9*, *Table 2*). Simulations predicted that stepwise increases in the rate of movement should progressively reduce branching (*Figure 9B,C*, *Videos 11, 12, 13*), and in DDG treated plants branching was similarly reduced (*Figure 9D–M*). These data suggest callose-gated plasmodesmal connectivity as a plausible mechanism to regulate branching patterns in *Physcomitrella*.

## Strigolactone fulfils a predicted requirement for basal suppression of branching

A final testable assumption of our model is that branching is locally suppressed at the base of the gametophore, expressed as a local reduction in *T* in our model (*Figure 3F*). We noted from published literature (*Proust et al., 2011*) that the strigolactone biosynthesis gene *ppccd8* is expressed at the base of gametophores. *ppccd8* mutants are strigolactone deficient and although no branching defects were previously detected in mutant gametophores (*Proust et al., 2011*), we reasoned that strigolactones could be the repressive signal. We therefore quantified the branching patterns in *ppccd8* mutant gametophores relative to WT (*Figure 10*). We found strongly increased branch activation at the base of gametophores (number of branching gametophores: WT, 4/60; *ppccd8* mutant, 29/60) (*Figure 10A,B*). To confirm that the *ppccd8* mutant phenotype was caused by a deficiency in strigolactone production, we quantified the branching pattern in *ppccd8* mutants grown on medium supplemented with 1 µM GR24, a synthetic strigolactone analogue, and found that GR24 was able to suppress basal branch formation (*Figure 10C*). To evaluate whether strigolactone can act as a global branch suppressor, we analysed the branching pattern in *Physcomitrella* mutants that over express the pea strigolactone synthesis gene, *RMS1* (*Proust et al., 2011*). We found that branching was very strongly suppressed in this transgenic line (*Figure 10D*, *Figure 10—figure supplement 1*). We used gametophores with fewer than 20 leaves for this experiment because comparison of model predictions (*Figure 3E,F*) to WT data (*Figure 2C*) showed that we should most clearly detect a difference at this stage of development, and *ppccd8* mutants make few gametophores. Our inferences were supported in a similar but scaled down experiment using fully grown gametophores (*Figure 10—figure supplement 1*). Thus basal expression of a strigolactone biosynthetic gene can account for the predicted requirement of our model for a local basal inhibitor of branching.

## Discussion

Hormonal signalling by auxin, cytokinin and strigolactone evolved before plants' transition to land (*Cooke et al., 2002*; *De Smet et al., 2011*; *Delaux et al., 2013*; *Gruhn and Heyl, 2013*), and was recruited to regulate axillary bud initiation (auxin and cytokinin; [*Wang et al., 2014a*, *2014b*]) and

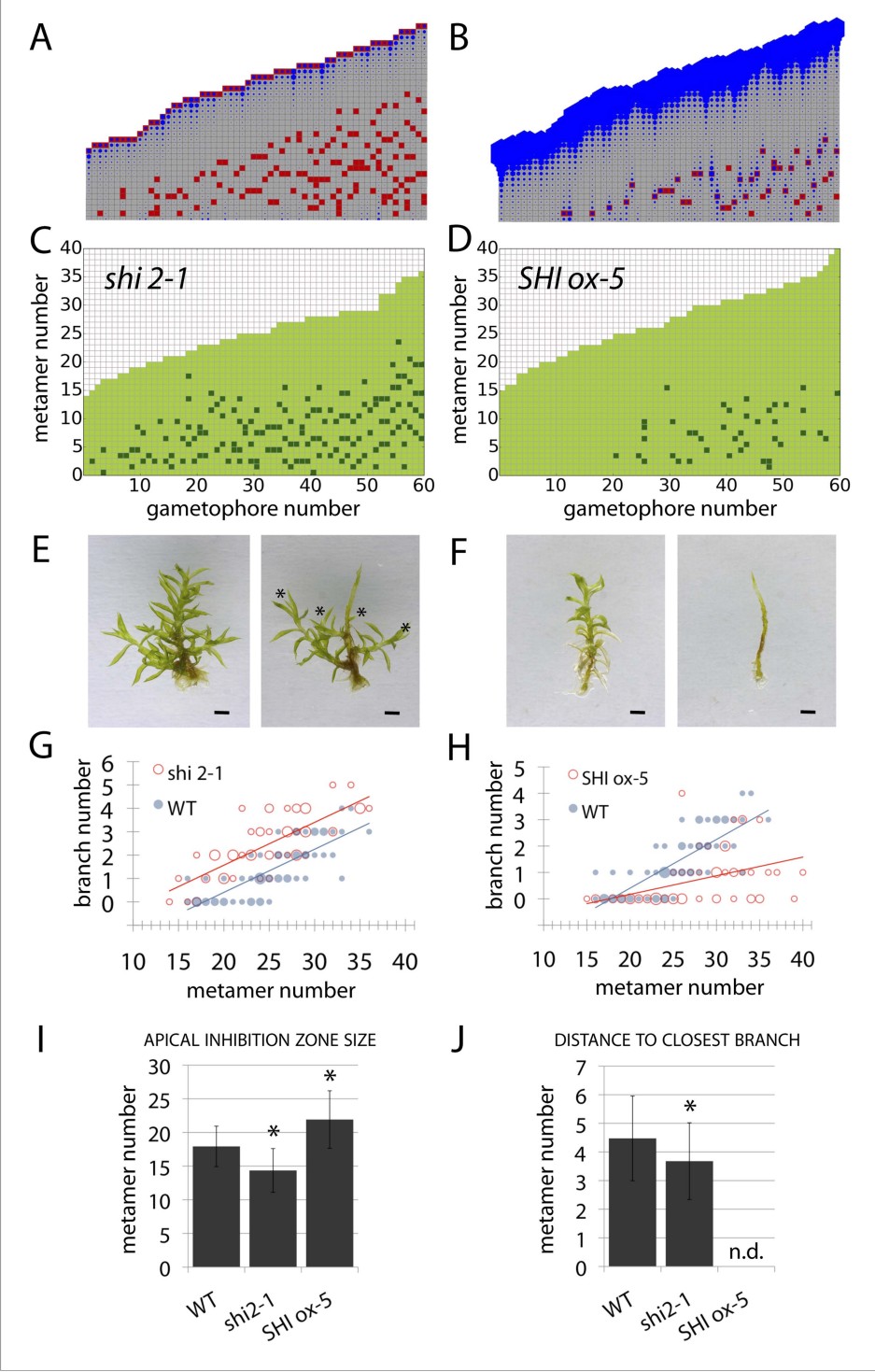

**Figure 6.** Auxin biosynthesis mutants and transgenics match predicted effects of changing parameter values of $H_{apex}$ and $H$. (**A–B**) Model simulations of branching patterns with $H_{apex}$ and $H$ values reduced from $80 \pm 20$ to $48 \pm 12$ and from $20 \pm 4.5$ to $12 \pm 2.7$ respectively (**A**) or $H_{apex}$ and $H$ values increased from $80 \pm 20$ to $240 \pm 60$ and from $20 \pm 4.5$ to $60 \pm 13.5$ (**B**). (**C–D**) Branching patterns in *shi2-1* mutants with reduced auxin biosynthesis levels (**C**) and *SHI ox-5* transgenics with elevated auxin biosynthesis levels (**D**). (**E–F**) *shi2-1* (**E**) and *SHI ox-5* (**F**) gametophores before (left) and after (right) removing the leaves, with asterisks indicating lateral branches. Scale bar = 1 mm. (**G–H**) Bubble plots showed that branch number increased in *shi2-1* (**G**) and diminished in *SHI ox-5* (**H**) compared with WT gametophores. Gametophore length is depicted as the number of metamers and the bubble area is proportional to

*Figure 6. continued on next page*

*Figure 6. Continued*

the number of gametophores with a similar branch number at a particular length. For (**G**) the best fitting model was B = (−3.27 + 1.16X) + 0.18L; *shi2-1* significantly differed from WT (p < 0.001). For (**H**) the best fitting model was B = (−3.29 + 2.05X) + (0.18 − 0.11X)L, *SHI ox-5* significantly differed from WT (p < 0.001). (**I**) Apical inhibition zone size was reduced in *shi2-1* and increased in *SHI ox-5* (mean ± SD; bilateral t-test different from WT, *p < 0.05). (**J**) Minimum distance between lateral branches was reduced in *shi2-1* (mean ± SD; bilateral t-test different from WT, *p < 0.05). n.d., not determined because branch number was insufficient.

The following figure supplement is available for figure 6:

**Figure supplement 1**. Comparison of mutant branching pattern plots with model outputs.

lateral branch outgrowth in flowering plant sporophytes (*Domagalska and Leyser, 2011*). We have shown that these three ancient hormonal cues were recruited independently to regulate lateral branching in gametophores of the moss *Physcomitrella*. In combination our model and experimental data lead us to a notion of branching whereby the ratio *c/T* determines the fate of epidermal cells in the gametophore axis. If the ratio drops below a threshold level, an epidermal cell can be respecified as a lateral apical cell thereby triggering branch initiation. Such a drop usually occurs in leaf axils independently of the leaf initiation process at the meristem (*Harrison et al., 2009*) as a result of competing hormonal cues dispersed across the gametophore. However, cells elsewhere in the metamer are also competent to form branch initials, and do so if the *c/T* ratio is perturbed, for instance by changing auxin (*Figures 5, 6*), strigolactone (*Figure 9*) or cytokinin levels (*Figure 7*). Although this process has some similarities to the branch initiation process in *Arabidopsis* in which a drop in auxin levels in the axils of initiating leaves coupled with a rise in cytokinin levels primes the initiation of the axillary meristem (*Wang et al., 2014a, 2014b*), there are many differences in the regulation of lateral branching between flowering plant sporophytes and *Physcomitrella* gametophytes.

A first key difference is the mechanism underlying apical dominance, which requires PIN-mediated bulk basipetal polar auxin transport in *Arabidopsis* (*Gälweiler et al., 1998*; *Ljung et al., 2001*).

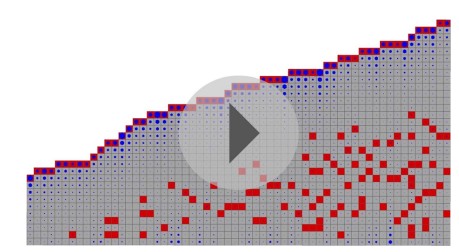

**Video 7.** (corresponds to *Figure 6A*) Simulation of branching activation pattern in the *shi2-1* mutant.

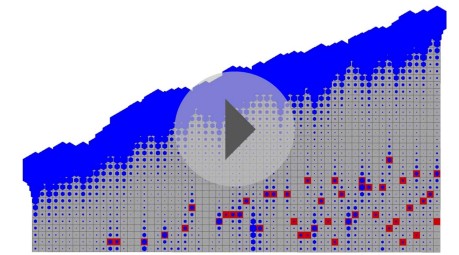

**Video 8.** (corresponds to *Figure 6B*) Simulation of branching activation pattern in the *SHI ox-5* transgenic line.

Our model for branching in *Physcomitrella* assumes that the main gametophore apex acts as the source of a cue that can move into the gametophore axis to suppress branching. Decapitation experiments show that the main gametophore apex inhibits branching and that this inhibitory effect can be maintained if the apex is substituted by a source of auxin (*Figure 4*). We detect higher levels of auxin in the apex than elsewhere in the gametophore, and removal of the apex diminishes levels of expression of an auxin responsive reporter in the gametophore axis, thereby implicating auxin as an apical cue and indicating a requirement for auxin transport away from the apex in the regulation of branching patterns (*Figure 4*). Although these results are qualitatively similar to the outcome of such experiments in *Arabidopsis* (*Thimann and Skoog, 1933*; *Wickson and Thimann, 1958*; *Cline, 1991*), no bulk basipetal auxin transport has been detected in *Physcomitrella* (*Fujita et al., 2008*). Our model predicts a requirement for bi-directional, or diffusion-like transport to generate realistic patterns (*Figure 3*), and suggests that not only the direction (*Figure 3*), but also the rate of transport has a significant impact on branch patterning

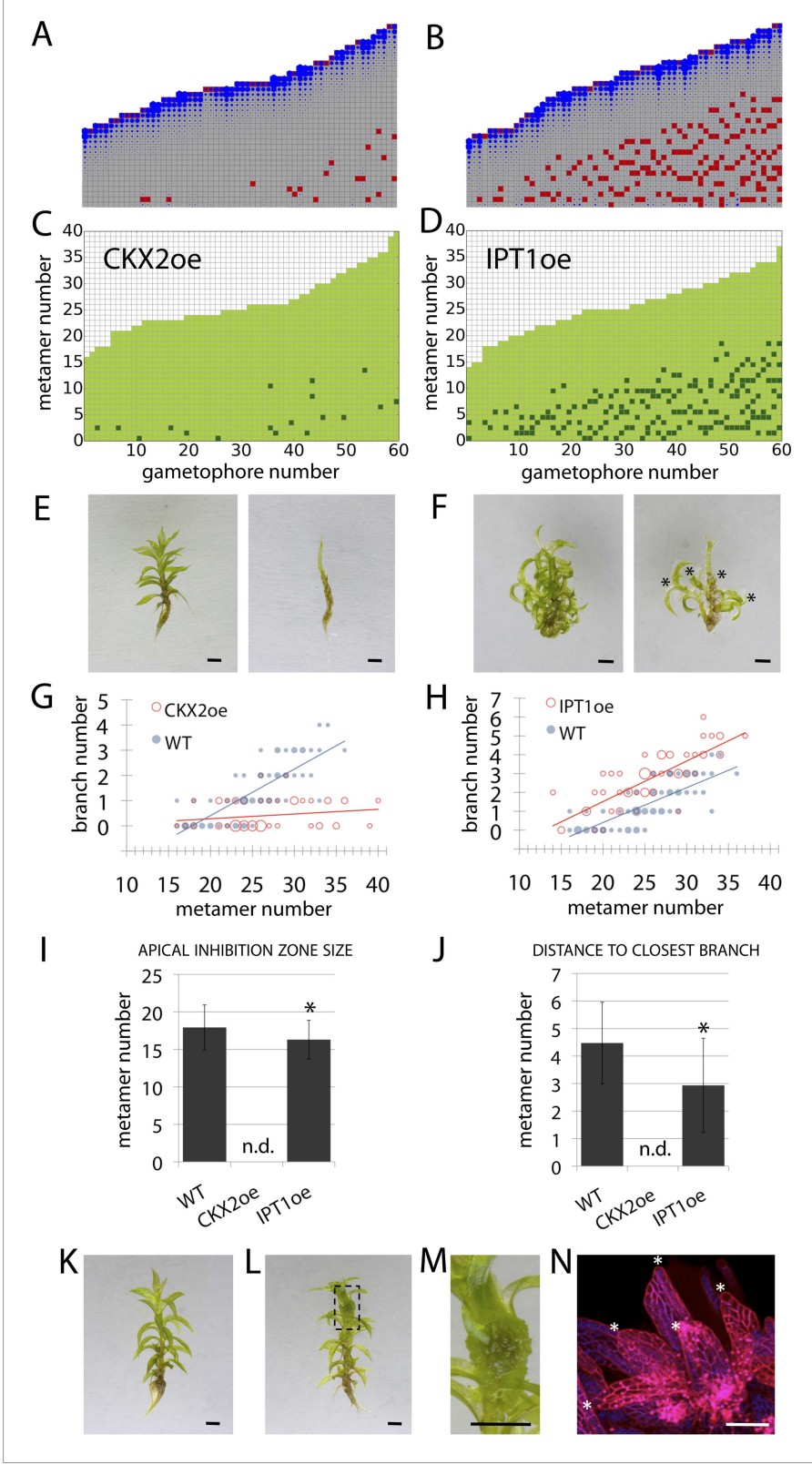

**Figure 7**. Cytokinin biosynthesis mutants and transgenics match predicted effects of changing parameter values of *T* and exogenous cytokinin treatment is sufficient for lateral meristem formation. (**A**) Model simulation of branching patterns with *T* values reduced from 3 ± 0.8 to 0.45 ± 0.2. (**B**) Model simulation of branching patterns with *T* values

*Figure 7. continued on next page*

*Figure 7. Continued*

increased from 3 ± 0.8 to 5.4 ± 0.8. (**C–D**) Branching patterns in *CKX2oe* transgenics with reduced cytokinin levels (**C**) and *IPToe* transgenics with increased cytokinin levels (**D**). (**E–F**) *CKX2oe* (**E**) and *IPToe* (**F**) gametophores before (left) and after (right) removing the leaves, with asterisks indicating lateral branches. Scale bar = 1 mm. (**G–H**) Bubble plots showed that branch number diminished in *CKX2oe* (**G**) and increased in *IPToe* (**H**) compared with WT gametophores. Gametophore length is depicted as the number of metamers and the bubble area is proportional to the number of gametophores with a similar branch number at a particular length. (**G**) The best fitting model was B = (−3.28 + 3.19X) + (0.18 − 0.17X)L, *CKX2oe* significantly differed from WT (p < 0.001). (**H**) The best fitting model was B = (−3.69 + 1.28X) + 0.2L, *IPToe* significantly differed from WT (p < 0.001). (**I**) Apical inhibition zone size was reduced in *IPT1oe* (mean ± SD; bilateral t-test different from WT, *p < 0.05). (**J**) Minimum distance between lateral branches was reduced in *IPT1oe* (mean ± SD; bilateral t-test different from WT, *p < 0.05). n.d., not determined because branch number was insufficient. (**K–N**) Exogenous cytokinin treatment promoted branch initiation. WT gametophores 1 week after immersion in mock (**K**) or 1 μM BAP (**L**) solution for 24 hr. Cytokinin promoted development of callus-like structures (**M**). Scale bar = 1 mm. (**N**) Confocal microscope image showing that cytokinin-induced structures were mainly constituted of leaves (white asterisks) resulting from the proliferation of ectopic meristematic cells. Scale bar = 100 μm.

The following figure supplement is available for figure 7:

**Figure supplement 1**. Molecular characterization and cytokinin profiling of the *PpIPT1* overexpressing line.

(*Figure 9*). Our experiments with *pin* mutants and pharmacological inhibitors show that membrane-targeted auxin transporters belonging to PIN and ABCB/PGP families are not significant contributors to branch patterning (*Figure 8*). Patterning is sensitive to the application of the callose synthesis inhibitor DDG, and incremental increases in the concentration of applied DDG have similar effects on branch patterning as incremental increases in the rates of transport implemented in our model (*Figure 9*). These data suggest that auxin may move with diffusive properties via callose-gated plasmodesmatal connections between cells (*Han et al., 2014*).

Further distinctions relate to variation in conceptualized notions of global and local sensitivities to auxin that can be produced in moss by varying the activity of cytokinin biosynthesis and degradation, and strigolactone biosynthesis pathways respectively. Although the role of cytokinin in gametophore induction is well characterized in *Physcomitrella* (*Ashton et al., 1979*), and cytokinin has previously been shown to antagonize auxin in branch formation (*von Maltzahn, 1959*), little is known about where and how cytokinin acts later in gametophore development. Our results are consistent with a homogenous effect of cytokinin within the gametophore axis (*Figure 8*), and further analysis of cytokinin distributions, for instance with a modified TCS reporter (*Müller and Sheen, 2008*), may be informative. Upregulation of cytokinin levels by overexpressing *IPT1*, and exogenous cytokinin applications show that cytokinin is sufficient to promote lateral meristem formation and therefore indicate that cytokinin directly promotes branching (*Figure 7*).

The mechanism by which strigolactone acts with auxin in the regulation of branching is likely to differ between *Arabidopsis* and *Physcomitrella*. Whereas strigolactone is thought to suppress branch outgrowth in *Arabidopsis* by dampening PIN mediated polar auxin

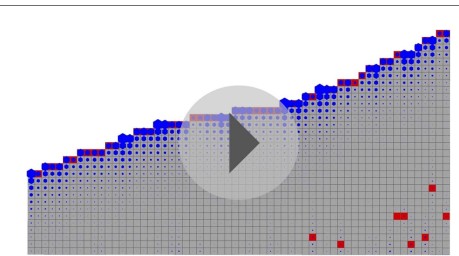

**Video 9.** (corresponds to *Figure 7A*) Simulation of branching activation pattern in the *CKX2oe* transgenic line.

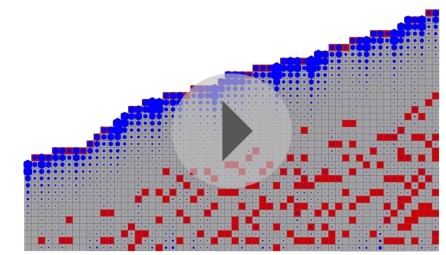

**Video 10.** (corresponds to *Figure 7B*) Simulation of the branching activiation pattern in the *IPT1oe* transgenic line.

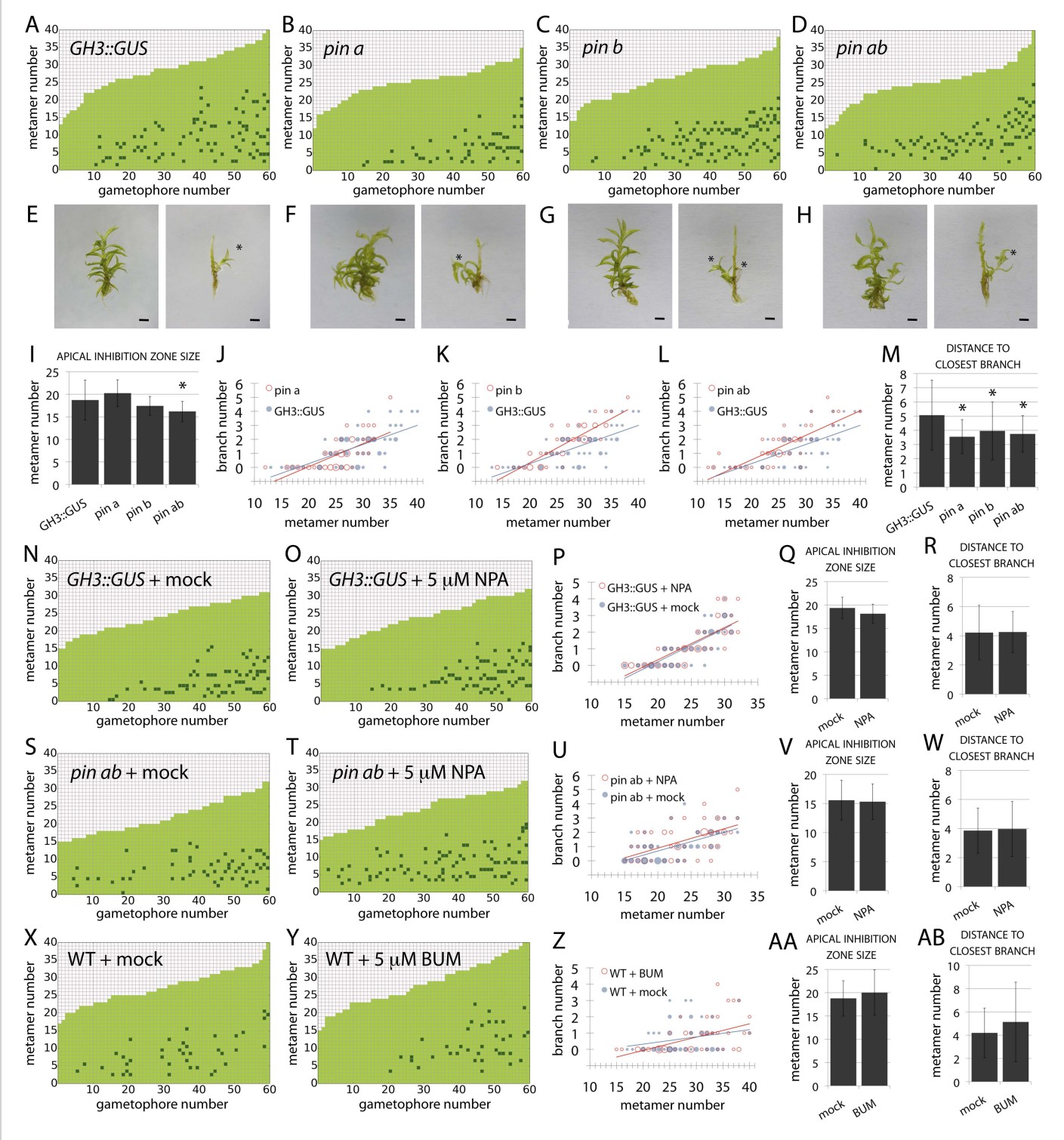

**Figure 8**. PIN-mediated auxin transport is a minor contributor to branching patterns. (**A–D**) Branching patterns in *GH3::GUS*, *pina*, *pinb* and *pina pinb* mutants. (**E–H**) *GH3::GUS* (**E**), *pina* (**F**), *pinb* (**G**) and *pina pinb* (**H**) gametophores before (left) and after (right) removing leaves, with asterisks indicating lateral branches. Scale bar = 1 mm. (**I**) Apical inhibition zone size is significantly reduced in the *pina pinb* double mutant but not in *pina* or *pinb* single mutants (mean ± SD; bilateral t-test different from *GH3::GUS*, *p < 0.05). (**J–L**) Bubble plots showed that branch number increases in *pinb* (**K**) and *pina pinb* (**L**) but not *pina* (**J**) compared with *GH3::GUS* gametophores. Gametophore length is depicted as the number of metamers and the bubble area is proportional to the number of gametophores with a similar branch number at a particular length. The best fitting model for (**J**) was B = −2.8 + 0.15L, *pina* was not significantly different from *GH3::GUS*. The best fitting model for (**K**) was B = (−2.47 − 1.40X) + (0.14 + 0.07X)L, *pinb* was significantly different from

*Figure 8. continued on next page*

*Figure 8. Continued*

GH3::GUS (p < 0.01). The best fitting model for (**L**) was B = (−2.96 + 0.52X) + 0.15L, *pina pinb* significantly differed from *GH3::GUS* (p < 0.01). (**M**) The minimum distance between lateral branches is reduced in *pina*, *pinb* and *pina pinb* mutants with respect to *GH3::GUS* controls (mean ± SD; bilateral t-test different from *GH3::GUS*, *p < 0.05), thus branch density in the branching zone is higher in all the mutants. (**N–O**) Branching patterns in *GH3::GUS* mutants treated without (**N**) or with (**O**) 5 μM NPA. (**P–R**) 5 μM NPA treatment did not affect the branch number in *GH3::GUS* transgenics (**P**), the apical inhibition zone size (**Q**) and the minimum distance between lateral branches (**R**). For (**P**), the best fitting model was B = −3.64 + 0.2L; NPA treatment was not significantly different from the mock treatment. (**S–T**) Branching patterns in *pina pinb* mutants treated without (**S**) or with (**T**) 5 μM NPA. (**U–W**) 5 μM NPA treatment did not affect the branch number in *pina pinb* mutants (**U**), the apical inhibition zone size (**V**) and the minimum distance between lateral branches (**W**). For (**U**), the best fitting model was B = −1.99 + 0.14L; NPA treatment was not significantly different from the mock treatment. (**X–Y**) Branching patterns in WT treated without (**X**) or with (**Y**) 5 μM BUM. (**Z–AB**) 5 μM BUM treatment did not affect the branch number in WT (**Z**), the apical inhibition zone size (**AA**) and the minimum distance between lateral branches (**AB**) (mean ± SD). For (**Z**), the best fitting model was B = −1.24 + 0.07L; BUM treatment was not significantly different from the mock treatment.

transport, the mode of action of strigolactone in *Physcomitrella* is unclear. Notably, key strigolactone signalling components required for the inhibition of sporophyte branching in higher plants are not present in mosses (*Delaux et al., 2013*; *de Saint Germain et al., 2013*). For instance, components such as D14 (involved in strigolactone perception) and D53 are absent in *Physcomitrella* (*Challis et al., 2013*; *Delaux et al., 2013*; *Zhou et al., 2013*). Similarly the *Physcomitrella* homologue of the F-Box protein MAX2 may not be involved in strigolactone signalling, despite its central role in many strigolactone responses in flowering plants (*de Saint Germain et al., 2013*). The modulation of branching by *PpCCD8* and *RMS1* (*Figure 10*, *Figure 10—figure supplement 1*) was unexpected as previously published work identified roles for *PpCCD8* in protonemal but not gametophore branching patterns (*Proust et al., 2011*). The basal expression domain of *PpCCD8* (*Proust et al., 2011*) is consistent with a local site of hormone action, but the mechanism underlying the *ppccd8* and *RMS1oe* mutant phenotypes remains to be seen. Although there is a substantial body of evidence that strigolactone-like signals are active across the plant kingdom, the ancestral function of strigolactone and/or these strigolactone-like signals is thought to be in rhizoid elongation (*Delaux et al., 2013*; *de Saint Germain et al., 2013*). The absence of distinct branch initiation and outgrowth processes in *Physcomitrella* implicates strigolactone in branch initiation, and the relative unimportance of PIN-mediated auxin transport in the regulation of *Physcomitrella* branching patterns suggests that strigolactone is unlikely to act via PIN-mediated auxin transport.

Diverse branching forms are evident in both the gametophytic (*Taylor et al., 2005*) and sporophytic plant fossil record (*Gerrienne et al., 2006*; *Edwards et al., 2014*). Although plant phylogenies show that gametophytic branching forms have several independent evolutionary origins, the innovation of sporophytic branching maps once onto the plant tree of life (*Langdale and Harrison, 2008*) and occurred in pre-vascular plants such as *Partitatheca* (*Edwards et al., 2014*). Exogenous hormone treatments in *Selaginella* have shown that the antagonistic relationship between auxin and cytokinin in the regulation of branching is conserved within the vascular plants (*Sanders and Langdale, 2013*). Furthermore, auxin transport assays and pharmacological treatments used in combination with decapitation in *Selaginella* (*Williams, 1937*; *Wochok and Sussex, 1973*, *1975*) have shown that auxin transport-mediated apical dominance is a homology of vascular plant sporophytes. Disruption of PIN-mediated polar auxin transport in moss sporophytes can induce a branching form that is intermediate between living bryophytes and vascular plants (*Fujita et al., 2008*; *Bennett et al., 2014b*), and resembles the earliest branching forms in the sporophytic fossil record (*Edwards et al., 2014*), suggesting that auxin transport-regulated branching may be a homology of the sporophyte generation. Although the regulation of lateral branching by three key regulatory hormone pathways in moss gametophytes indicates deep homology (*Scotland, 2010*), the mode and basis of auxin transport are a key divergence point in branch patterning mechanisms between land plant sporophytes and gametophytes.

## Materials and methods

Sterile spot cultures were grown on BCD + AT medium as described previously (*Harrison et al., 2009*) and gametophores were sampled after 5 and 7 weeks for branching pattern analysis. For histological analyses, gametophores from 6-week-old colonies were fixed in 1% paraformaldehyde, 3% glutaradehyde, 0.5% tannic acid in phosphate buffered saline overnight at 4°C. Fixed tissue was

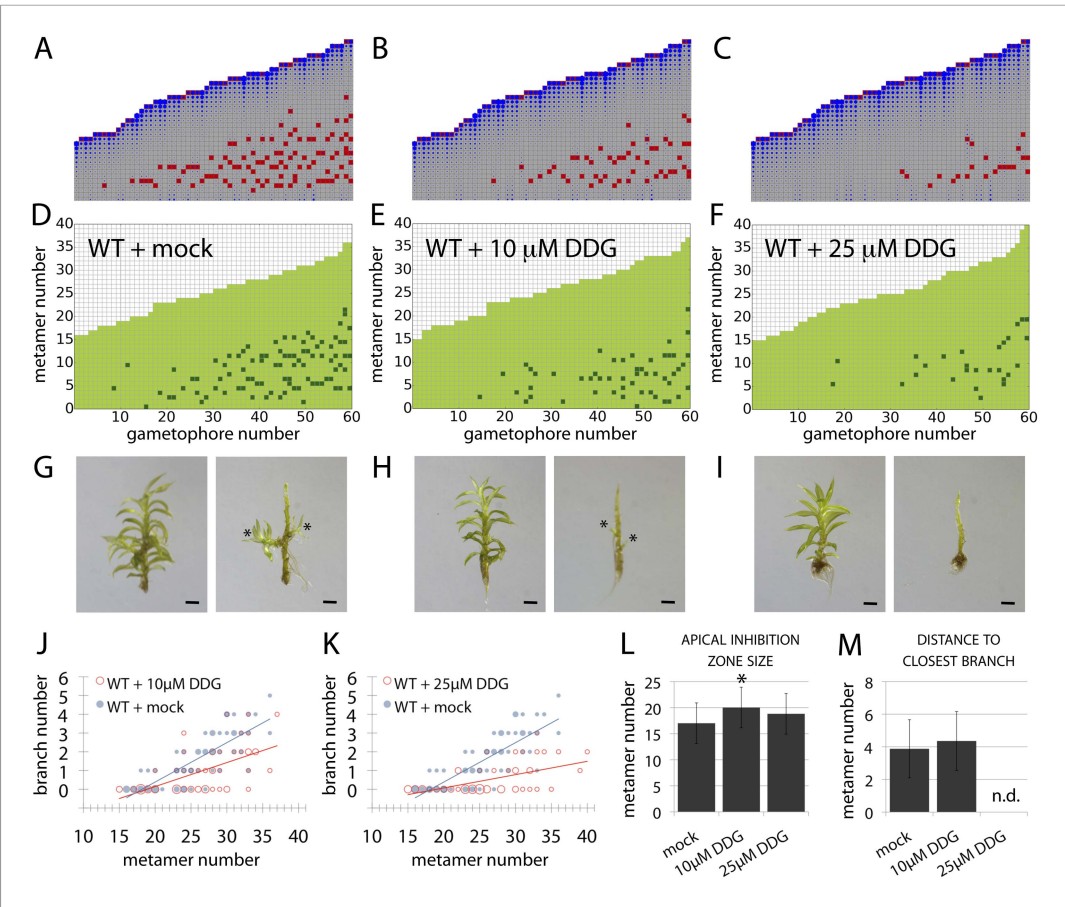

**Figure 9**. A callose synthesis inhibitor regulates branching. (**A–C**) Model simulation of branching patterns with *K* values increased from 0.025 (**A**) to 0.045 (**B**) and 0.07 (**C**). (**D–F**) Branching patterns in wild-type gametophores treated with mock (**D**), 10 μM 2-deoxy-glucose (DDG) (**E**) or 25 μM DDG (**F**). (**G–I**) mock (**G**), 10 μM DDG (**H**) or 25 μM DDG (**I**) treated gametophores before (left) and after (right) removing the leaves, with asterisks indicating lateral branches. Scale bar = 1 mm. (**J–K**) Bubble plots show that branch number diminishes in 10 μM DDG (**J**) and is strongly reduced in 25 μM DDG (**K**) treated gametophores compared with mock treatment. Gametophore length is depicted as the number of metamers and the bubble area is proportional to the number of gametophores with a similar branch number at a particular length. The best fitting model for (**J**) was B = (−3.82 + 1.41X) + (0.21 − 0.08X) L, 10 μM DDG was significantly different from mock treatment (p < 0.001). The best fitting model for (**K**) was B = (−3.82 + 2.47X) + (0.21 − 0.14X)L, 25 μM DDG was significantly different from mock treatment (p < 0.001). (**L**) Apical inhibition size is significantly increased in response to DDG treatment (mean ± SD; bilateral t-test different from WT mock, *p < 0.05). (**M**) The minimum distance between lateral branches is not different in 10 μM DDG treated gametophores with respect to mock controls (mean ± SD). n.d., not determined because branch number was insufficient.

The following figure supplement is available for figure 9:

**Figure supplement 1**. Comparison of branching pattern plots from pharmacologically treated plants with model outputs.

dehydrated and embedded in Technovit 7100 resin as described by the manufacturer (Kulzer, Germany). Five micrometer sections were cut and stained with 0.1% Toluidine Blue for 30 s, rinsed in water, and dried for storage or further examination. For SEM, samples were frozen from fresh on a Pelletier cooling stage set to −25°C and imaged using a Zeiss EVO HD (Zeiss, Germany) environmental scanning electron microscope with variable pressure mode and 30 Pa nitrogen gas. Models were implemented as in the SI methods. For analysis of endogenous IAA and CK content, gametophores were isolated from 6-week old colonies, dissected to collect 10–20 mg (for IAA) and

**Table 2**. Refitted parameter values for the models shown in **Figure 9**

|  | Model wild-type | Model 10 µM DDG | Model 25 µM DDG |
|---|---|---|---|
|  | (*Figure 9A*) | (*Figure 9B*) | (*Figure 9C*) |
| $H_{apex}$ | $\mu = 30$, $\sigma = 10$ |  |  |
| $H$ | $\mu = 5$, $\sigma = 1.5$ |  |  |
| $T$ | $\mu = 0.7$, $\sigma = 0.2$ |  |  |
| $\nu$ | 0.002 |  |  |
| $K_A$ | 0.025 | 0.045 | 0.07 |
| $K_B$ | 0.025 | 0.045 | 0.07 |

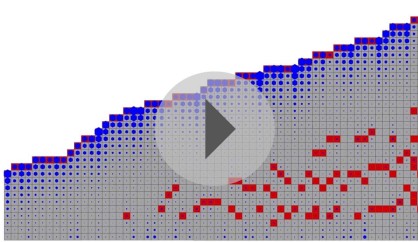

**Video 11.** (corresponds to *Figure 9A*) Simulation of the WT branching activation pattern with refitted parameters.

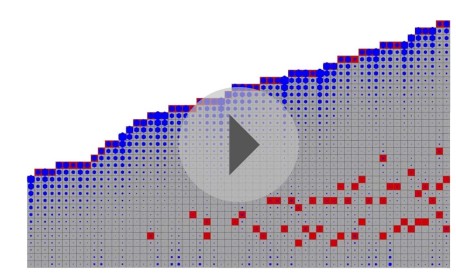

**Video 12.** (corresponds to *Figure 9B*) Simulation of branching activation pattern with *K* values increased by 100%.

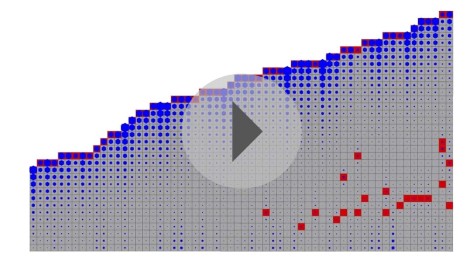

**Video 13.** (corresponds to *Figure 9C*) Simulation of branching activation pattern with *K* values increased by 200%.

50 mg (for CK) of tissue and snap frozen in liquid nitrogen. Samples were extracted, purified and analyzed using gas chromatography-tandem mass spectrometry as previously described, *Edlund et al. (1995)* for IAA and *Novák et al. (2008)* for CK. Each measurement is the mean of four (CK) or five (IAA) biological replicates.

## Model implementation

All computational models were created using the VVe modeling environment (*Abley et al., 2013*). The moss gametophore is represented in the model as a graph of vertices and connecting edges. Each vertex represents either a metamer or an apex. The graph representing the moss is dynamic. Periodically, new vertices are added to the graph representing growth of the gametophore (*Figure 2G*). The number of simulation steps between the addition of new vertices in the model constitutes 1 plastochron, which we assume to be approximately 1 day for wild-type *Physcomitrella patens* grown on agar medium. The differential equations calculating the rate of change of auxin concentration per vertex (*Equation 1*) of the graph are solved numerically using the forward Euler integration scheme. The time step Δt is set to 0.005, there are thus 200 simulation steps per plastochron. For all simulation results presented, a number of gametophores have been simulated where growth is delayed by a random amount of simulation steps between two neighboring axes, and the simulation stops after a set number of total simulation steps has been reached. All simulated gametophores initiate as one apical vertex visualized in red, and a vertex representing the first metamer, visualized in grey. Separate lateral axes are simulated, and their initiation point is represented as a red cell. Higher order branching is not represented and was rarely observed in experiments.

eLIFE Research article

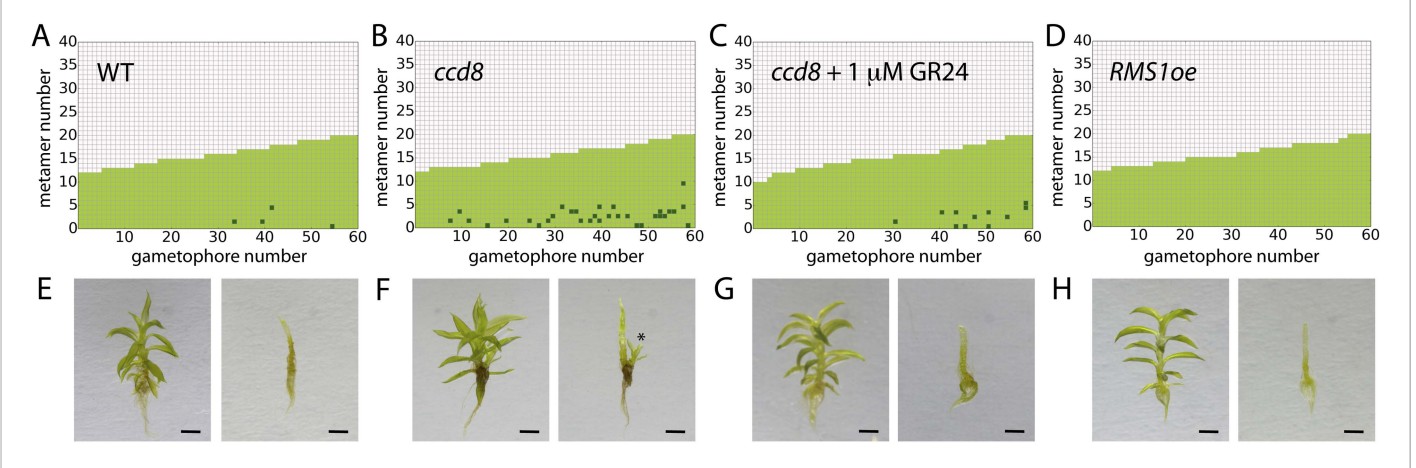

**Figure 10**. Expression levels of a strigolactone biosynthesis gene modulates branching. (**A–D**) Branching patterns of 60 gametophores with fewer than 20 leaves in (**A**) WT, (**B**) strigolactone-deficient *ppccd8* mutants, (**C**) *ppccd8* mutants treated with 1 µM GR24 and (**D**) *RMS1oe* transgenics with increased strigolactone levels. (**E–H**) WT (**E**), *ppccd8* (**F**), GR24-treated *ppccd8* (**G**) and *RMS1oe* (**H**) gametophores before (left) and after (right) removing the leaves, with asterisks indicating lateral branches. Scale bar = 1 mm.

The following figure supplement is available for figure 10:

**Figure supplement 1**. Branching patterns of gametophores with more than 20 leaves and branch proportion in the five most basal metamers.

## Model parameter values

For the most important parameters such as auxin concentrations, auxin movement or auxin decay rates there were no direct estimates available to this project. Nevertheless, we assume that auxin movement in the gametophore of *Physcomitrella* is slower than in *Arabidopsis* shoot tissue (~20 mm/hr) and faster than in NPA treated *Arabidopsis* shoot tissue (~0.1 mm/hr) choosing an arbitrary value inside these bounds for our simulations. Our parameter value settings, given that we have 200 simulation steps per plastochron, correspond to an auxin movement rate of approximately 1 mm/hr in the gametophore. However, we found for several auxin movement rate values inside these bounds for which simulation results could be reproduced by readjusting other parameter values. For example, the simulations represented in *Figure 9* use a smaller time step $\Delta t$ of 0.001 compared to all other simulations.

## Statistical analysis

Ordinary least squares regression was used to test whether the relationship between branch number and gametophore leaf number depended on genotype. A linear model $B = (a_0 + a_1 X) + (a_2 + a_3 X) L$, where $a_0$, $a_1$, $a_2$ and $a_3$ are coefficients, $L$ is the number of leaves, $B$ the number of branches, and $X$ is an indicator variable depending on genotype (0 corresponded to a mutant, 1 corresponded to WT) was fitted. Backwards stepwise elimination was used to find the minimal model with support from the experimental data. A mutant was considered as different from control genotype if either the interaction term ($a_3$) or the term corresponding to genotype ($a_1$) remained in the minimal model.

## Acknowledgements

We thank Catherine Rameau, Eva Sundberg and Klaus von Schwartzenberg for giving us mutant lines, Nik Cunniffe for his support with statistical analyses and Siobhan Braybrook for help with the scanning electron microscope. We thank our funding bodies for financial support. Yoan Coudert and Jill Harrison are funded by a BBSRC grant 'PIN proteins and architectural diversification in plants' (Grant BB/L00224811) and fellowships from the Gatsby Charitable Foundation (GAT2962) and Royal Society. Ottoline Leyser and Wojtek Palubicki are funded by the Gatsby Charitable Foundation

(Grant GAT3272C) and by the European Research Council (Grant N° 294514—EnCoDe). Karin Ljung is funded by the Swedish Governmental Agency for Innovation Systems (VINNOVA) and the Swedish Research Council (VR) and thanks Roger Granbom for excellent technical assistance. Ondrej Novak is funded by a Czech Ministry of Education grant from the National Program for Sustainability I (LO1204).

## Additional information

### Funding

| Funder | Grant reference | Author |
|---|---|---|
| Gatsby Charitable Foundation | | Yoan Coudert, C Jill Harrison |
| Biotechnology and Biological Sciences Research Council (BBSRC) | BB/L00224811 | Yoan Coudert, C Jill Harrison |
| Royal Society | | Yoan Coudert, C Jill Harrison |
| European Research Council (ERC) | 294514-EnCoDe | Wojtek Palubicki, Ottoline Leyser |
| VINNOVA | | Karin Ljung |
| Swedish Research Council Formas (Svenska Forskningsrådet Formas) | | Karin Ljung |
| Ministry of Education, Youth and Sports | LO1204 | Ondrej Novak |
| Gatsby Charitable Foundation | GAT3272C | Wojtek Palubicki, Ottoline Leyser |

The funders had no role in study design, data collection and interpretation, or the decision to submit the work for publication.

### Author contributions

YC, WP, Conception and design, Acquisition of data, Analysis and interpretation of data, Drafting or revising the article; KL, ON, Conception and design, Acquisition of data, Analysis and interpretation of data; OL, CJH, Conception and design, Analysis and interpretation of data, Drafting or revising the article

### Author ORCIDs

Karin Ljung, http://orcid.org/0000-0003-2901-189X

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
