## [Decision Letter]

Thank you for choosing to send your work entitled “Three ancient hormonal cues co-ordinate shoot branching in a moss” for consideration at *eLife*. Your full submission has been evaluated by Detlef Weigel (Senior editor), a Reviewing board editor and three peer reviewers, and the decision was reached after discussions between them. Based on our discussions and the individual reviews below, we regret to inform you that your work will not be considered further for publication in *eLife* at this time.

As you can see from their attached reviews, the reviewers acknowledge your interesting comparative study of branching in seedless and seed plants, but they also feel that the story is not yet developed far enough. You will note that the reviewers raise some important criticism of the modelling part of your paper. Specifically, they point out discrepancies that apparently go against your statement that the model “accurately reproduces observed branching patterns”. Should you be able to come up with a substantially improved and further developed manuscript in the future, we would be happy to consider it, but please be aware that this would be handled as a new submission.

[Editors’ note: a new submission was re-reviewed and accepted without further revisions.]

*Reviewer #1*:

The authors study branching in *P. Patens* using a combination of morphological characterization, genetics, pharmacology and computational modeling. The subject area is interesting and a comparative understanding of branching in seedless and seed plants of value to the field. As such the study provides some potentially interesting hints on this front. However clear cut findings do not yet emerge so the case for publication in a general interest journal is not yet very strong. There are also technical issues that need resolution.

1) The initial simulations showed a branch formed always in the most basal metamer. Is there biological basis for this or is it a computational artefact relating to boundary conditions used? Figure 3 shows a lot of simulations that there is a gradient in auxin emanating from the base. Where does this auxin come from? This seems problematic. These issues need a very clear explanation.

2) The proposed mechanism does not seem very robust. What stops branches from emerging next to each other? Neighboring sites could be (randomly) assigned a very close T which would result in branches in neighboring metamers. This does seem to be the case in Figure 3 where there are 7 side-by-side initiations. If you compare with the branching pattern in Figure 2 you will there are none of these cases in the WT data. The long-range inhibition that is currently there (auxin diffuses from sources and branches are only made under a target concentration) is very unlikely be enough to account for this. What do the authors suggest? Is there a biological entry point into modeling these key aspects?

3) The treatment of lateral branches is not explained very clearly in the text. In the supplement the authors state they do grow out, so I assume their distance from the main axis would indeed affect their inhibitory effect. A cartoon or a model visualization of a typical run of the model including the branches would be helpful to understand this point. It is also not clear whether the distance to the lateral tips is taken into account when calculating distances between branches. If not, shouldn't it be? In general the relationship between the length of lateral branches and their inhibiting effect needs consideration.

4) On the same front: Why is it not possible to decapitate/ablate lateral branches? That experiment would really help modeling efforts.

5) While the overexpression of *RMS1* does appear to have a strong phenotype the role of strigolactone in the model is less clear. It seems the phenotype of *ppccd8* is very subtle and only detectable in plants with fewer than 20 leaves. If the basal suppression is strigolactone related, then why don't we see it in older plants? In the model, a strigolactone induced reduction of the threshold is required in the basal “5 or so” cells, yet there are often branches formed in these regions in real plants (see Figures 2 and 5).

6) The question of how auxin is transported seems very pertinent to the model and as far as I understand we do not yet have a clear picture on the biology here. Data on the *pin* mutants seems not fully conclusive but consistent with the PINs being apolar transporters, however the authors claim transporters play a minor role. If neither PINs nor the ABCBs are responsible for this transport, how does the auxin get from cell to cell? It cannot really diffuse across the membrane, yet the model suggests a diffusive mechanism, such as apolar transport. This makes the whole argument a bit confusing and weak. The reader is left feeling there are too many uncertainties piling up here to allow the model to provide substantial novel insight into the process.

7) The *GH3:GUS* expression in relation to auxin quantification is not clearly explained. How reliable an auxin reporter is it?

*Reviewer #2*:

The authors look at shoot branching in moss. They look at branch spacing and show that it appears not to be entirely random, and that there is some sort of spacing mechanism involved. They propose a model based on the interaction of auxin, cytokinin and strigolactone. The model consists of a growing apex that is a source of auxin, with the lateral branches also acting as auxin sources. Auxin diffuses away from the sources, and is degraded creating gradients. Cells that are below a threshold of auxin concentration initiate lateral branches. The concentration at sources is assigned with some random variation, as is the threshold for initiation in each cell. Cytokinin is added to the basic model by changing the threshold concentration of auxin below which outgrowth will occur, and strigolcatone is added by having an inhibitory effect at the base.

It is a very simple threshold model, which is also one of its major weaknesses. For example, in the model we see lateral initials in adjacent cells (Figure 3) which does not seem to occur in WT (Figure 2). Does this not suggest a process more active than a simple diffusion/threshold model, which is known not to be robust? The model needs some sort of local activation and long range inhibition, like a reaction-diffusion diffusion system, or like the model in Prusinkiewicz 2009, PNAS. It seems a bit difficult to compare distances in such a model with WT, when simple features of the distribution are different.

On the upside, the experimental work is nice as the authors have tested a variety of scenarios to see if the same players in branching in angiosperms are also used in moss. However the experiments leave us somewhat disappointed, as PINs and ABCBs appear not to be so important, but yet auxin still is. This begs the question as to how, exactly, cytokinin and strigolactone are thought to interact with auxin.

It would be nice to have more comparison throughout the text and results sections with the classic decapitation experiments in angiosperms, what outcomes are the same? What is different? Likewise for the auxin/cytokinin/strigolactone up/down manipulations.

Specific comments:

In the subsection headed “Strigolactone fulfils a predicted requirement for basal suppression of branching”: If the basal inhibitor in the model is indeed strigolactone, then why doesn't a branch always occur in the basal most metamer in the *ppccd8* mutant as in the model (compare Figure 9 with Figure 3)? The phenotype looks pretty weak compared to model predictions, suggesting that strigolactone must not be the main player for the basal repression.

Figure 3: Why does the cell on the bottom, 9th from the right, look to have a gradient of auxin coming from it, as though it was an auxin source?

*Reviewer #3*:

This paper demonstrates that auxin, cytokinin and strigolactones together coordinate branching in the gametophyte of *Physcomitrella*. This is important because it demonstrates that a similar mechanism has evolved to control the branching of bryophyte gametophytes and flowering plant sporophytes. However the authors identify some subtle differences.

The authors start with a model that indicates that a bidirectional transport is required to generate a pattern that reflects the pattern of branching along the moss gametophyte. They suggest that there are repressors and promoters of branching and then go onto identify regulatory molecules that fulfil the criteria predicted in the model.

Having constructed a model the authors then go on to test the biological significance of these parameters by experiment.

The first key experiment reports the finding that like sporophyte shoots, auxin produced at the apex represses branching. This was supported by the observation that auxin levels were higher at the apices suggesting that the apex is a source of auxin.

The authors then modulate endogenous levels of auxin using plants with decreased or increased levels of SHI2 activity. The authors state that branching pattern agrees with the predictions of the model. However they do not state the result and the reader is left to interpret the Figure 6 (see “suggestions” below). The figure indicates that loss of *shi2* function results in increased branching and over expression decreases branch number, which is consistent with their model. PIN mediated auxin transport is involved in branching development but is not the major component. This is an important result and explains published observations regarding auxin transport in gametophytes from the Hasebe group.

Cytokinin promotes branch formation and decreased endogenous levels of cytokinin results in a reduction in branching number.

Genetic evidence is presented that indicates that strigolactones repress branching.

Taken together the data support the model for branching. For the first time a coherent model that explains moss gametophore branching is presented.

Suggestions:

1) In the subsection headed “Auxin biosynthesis mutants capture predicted effects of changing model values H_apex_ and H”: This section is written cryptically. I don't have a problem with the data but the cryptic way in which the data are referred to. The authors should report the phenotype of *shi* mutants in the text. They only refer to H values, which is vague in the context of an experiment and experimental data. The final sentence of this section is unclear and vague—what is the biological result? The concluding sentence is no help to the reader (see helpful concluding sentence of the following paragraph). Even the title of Figure 6 is unhelpful and gnomic.

2) In the subsection headed “Up-regulating cytokinin degradation captures predicted effects of decreasing model values of *T*”, the expression “similar to the model output”. It would help a struggling reader if the authors stated the result and then concluded that his was similar to the model output.

---

## [Author Response]

We have explored changes in the rate of auxin transport via callose-gated plasmodesmatal connectvity as a plausible alternative mechanism in the regulation of branching patterns. We find that branch patterning is sensitive to the application of the callose synthesis inhibitor DDG, and incremental increases in the concentration of applied DDG have similar effects on branch patterning as incremental increases in the rates of transport implemented in model simulations.

We have substantially revised the manuscript to address further detailed reviewer comments as in the attached response, and hope that you will find our revised manuscript far enough developed to warrant publication.

Reviewer #1:

*The authors study branching in* P. Patens *using a combination of morphological characterization, genetics, pharmacology and computational modeling. The subject area is interesting and a comparative understanding of branching in seedless and seed plants of value to the field. As such the study provides some potentially interesting hints on this front. However clear cut findings do not yet emerge so the case for publication in a general interest journal is not yet very strong. There are also technical issues that need resolution*.

We think that our work makes a significant contribution to understanding branching in the moss gametophore, further strengthened by our addressing the helpful suggestions of the referees. In particular, recent publications from our and Professor Jiri Friml’s labs have shown that *Physcomitrella* PINs are auxin transporters that have many roles in gametophyte development, and our data showing that they do not regulate branching hold. We provide evidence that callose-gated plasmodesmatal connectivity is a plausible alternative mechanism in the regulation of branching patterns.

*1) The initial simulations showed a branch formed always in the most basal metamer. Is there biological basis for this or is it a computational artefact relating to boundary conditions used*?

The biological reason that branches initiate at the base is that auxin decay reduces the auxin concentration (calculated in each simulation step) below the threshold level of branch activation.

We have rewritten the main text explaining the model, and hope that this is now clear.

Figure 3
*shows a lot of simulations that there is a gradient in auxin emanating from the base. Where does this auxin come from? This seems problematic. These issues need a very clear explanation*.

The blue hexagons in Figure 3 represent the ratio between the auxin concentration (*c*) and the threshold for branch activation (*T*), as explained in Figure 2. In some simulated shoots in Figure 3, there is a high ratio of c/T because we have specified lower values of *T* in the basal metamers of the shoot. This means that the auxin concentration (*c*) does not drop below the threshold level for branch activation so readily. There is no auxin emanating from the base.

We have tried to clarify these aspects of the model by rewriting the explanatory text and changing the visualization to better distinguish between consideration of the auxin concentration (*c*), and *c/T*. We have also altered the visualization to show metamers in which the basal inhibitory cue is active (see ).

*2) The proposed mechanism does not seem very robust. What stops branches from emerging next to each other? Neighboring sites could be (randomly) assigned a very close T which would result in branches in neighboring metamers. This does seem to be the case in*
Figure 3
*where there are 7 side-by-side initiations. If you compare with the branching pattern in*
Figure 2
*you will there are none of these cases in the WT data. The long-range inhibition that is currently there (auxin diffuses from sources and branches are only made under a target concentration) is very unlikely be enough to account for this. What do the authors suggest? Is there a biological entry point into modeling these key aspects*?

We previously noted that branches do initiate next to each other during normal development (see Figures 4, 5 and 8) in the figure legend for Figure 2. Consequently, we see no discrepancy here between the model and the biological data. However, as this reviewer has sought to compare the model with data shown in Figure 2, we have revised the text explaining the model to highlight similarities.

*3) The treatment of lateral branches is not explained very clearly in the text. In the supplement the authors state they do grow out, so I assume their distance from the main axis would indeed affect their inhibitory effect. A cartoon or a model visualization of a typical run of the model including the branches would be helpful to understand this point. It is also not clear whether the distance to the lateral tips is taken into account when calculating distances between branches. If not, shouldn't it be? In general the relationship between the length of lateral branches and their inhibiting effect needs consideration*.

For the sake of simplicity in visualization, we abstracted the outgrowth of lateral branches to a single point on the stem, but it is correct that the distance of lateral branches from the main axis would affect their inhibitory effect.

We recognize that it might be helpful to visualize the progression of lateral branch development in the model, and have therefore included a simulation with lateral branch metamers in Figure 2.

*4) On the same front: Why is it not possible to decapitate/ablate lateral branches? That experiment would really help modeling efforts*.

We would like very much to be able to do these experiments. However, *Physcomitrella* gametophores are very small (5mm long) and branches initiate as a single cell. We have no genetic markers for branch initials, and cannot identify newly forming branches during growth.

*5) While the overexpression of* RMS1 *does appear to have a strong phenotype the role of strigolactone in the model is less clear*.

In the model, the basal inhibitor acts to increase the sensitivity to auxin by reducing the value of *T*. We have altered the model visualization to show metamers in which the basal inhibitory cue is active and rewritten the text explaining the model.

*It seems the phenotype of* ppccd8 *is very subtle and only detectable in plants with fewer than 20 leaves. If the basal suppression is strigolactone related, then why don't we see it in older plants*?

We have inserted a new figure in the supplement () showing that basal suppression is evident in older plants, but the phenotypes are clearer in younger shoots than in older shoots, so we have left the main text figure as was.

We have shown that upregulating strigolactone biosynthesis strongly antagonizes branch initiation, and that branch initiation is specifically upregulated at the base of *ppccd8* mutant gametophores. These data implicate strigolactones as important contributors to basal branching.

*In the model, a strigolactone induced reduction of the threshold is required in the basal* “*5 or so*” *cells, yet there are often branches formed in these regions in real plants (see*
Figures 2 and 5*)*.

The model visualization shown in Figure 3 illustrates the effect of a basal inhibitor on branch initiation, and shows that branches still initiate at the base of the gametophore. Therefore, we would not expect to see a difference in the frequency of branch initiation at the base between model runs and data from real plants. A comparison of the frequency of branches initiating in the basal 5 metamers of models shown in Figure 3, Figure 3 with WT datasets is shown in Figure 3–figure supplement 1C.

*6) The question of how auxin is transported seems very pertinent to the model and as far as I understand we do not yet have a clear picture on the biology here. Data on the* pin *mutants seems not fully conclusive but consistent with the PINs being apolar transporters, however the authors claim transporters play a minor role. If neither PINs nor the ABCBs are responsible for this transport, how does the auxin get from cell to cell? It cannot really diffuse across the membrane, yet the model suggests a diffusive mechanism, such as apolar transport. This makes the whole argument a bit confusing and weak. The reader is left feeling there are too many uncertainties piling up here to allow the model to provide substantial novel insight into the process*.

The data rule out major contributions to the regulation of branching by PINs or ABCBs, and point to an alternative mechanism of auxin transport in *Physcomitrella* gametophores. Diffusion-like properties are required to generate realistic branching patterns. Whilst we recognise cell to cell diffusion-like transport via the apoplast is unlikely without efflux carrier-mediated export, symplastic auxin transport via callose gated plasmodesmata has recently been shown to generate auxin gradients in *Arabidopsis* hypocotyls ([30]; Dev Cell), and we therefore tested whether this could be a plausible mechanism to account for the auxin transport required to regulate branching in the *Physcomitrella* gametophore.

We have undertaken experiments with callose synthesis inhibitors and have shown that the size of the apical inhibition zone and decreases the number of branches initiating relative to untreated controls, consistent with the notion of increased symplastic connectivity. We have included these data as a new figure (Figure 9) in the main text, and they fit the branch pattern predicted to arise in our model from an increase in the rate of global auxin transport.

*7) The* GH3:GUS *expression in relation to auxin quantification is not clearly explained. How reliable an auxin reporter is it*?

The *GH3:GUS* reporter is the only reporter for auxin distributions in *Physcomitrella* gametophores, and it is not yet known how accurately it reports the auxin distribution. Another project in our lab (Bennett et al. 2014; Current Biology) suggests that the reporter may not accurately reflect the auxin distribution in gametophore tips, which is why we attempted to quantify the auxin distribution. We have re-written the text to clarify these issues.

Reviewer #2:

*[…] It is a very simple threshold model, which is also one of its major weaknesses. For example, in the model we see lateral initials in adjacent cells (*Figure 3*) which does not seem to occur in WT (*Figure 2*). Does this not suggest a process more active than a simple diffusion/threshold model, which is known not to be robust? The model needs some sort of local activation and long range inhibition, like a reaction-diffusion diffusion system, or like the model in Prusinkiewicz 2009, PNAS. It seems a bit difficult to compare distances in such a model with WT, when simple features of the distribution are different*.

We previously noted that branches do initiate next to each other during normal development (see Figures 4, 5 and 8) in the figure legend for Figure 2. Consequently, we see no discrepancy here between the model and the biological data. As this reviewer has sought to compare the model with data shown in Figure 2, we have revised the text explaining the model to highlight similarities.

*On the upside, the experimental work is nice as the authors have tested a variety of scenarios to see if the same players in branching in angiosperms are also used in moss. However the experiments leave us somewhat disappointed, as PINs and ABCBs appear not to be so important, but yet auxin still is. This begs the question as to how, exactly, cytokinin and strigolactone are thought to interact with auxin*.

The data conclusively rule out major contributions to the regulation of branching by PINs or ABCBs, and point to an alternative mechanism of auxin transport in *Physcomitrella* gametophores. Auxin transport via a diffusion-like mechanism is required to generate realistic branching patterns.

Recent work in *Arabidopsis* has shown that symplastic auxin transport via callose gated plasmodesmata can generate auxin gradients in *Arabidopsis* hypocotyls ([30]; Dev Cell), and we therefore tested whether this could be a plausible mechanism to account for the auxin transport required to regulate branching in the *Physcomitrella* gametophore.

We have undertaken experiments with callose synthesis inhibitors and have shown that suppression of callose synthesis increases the size of the apical inhibition zone and decreases the number of branches initiating relative to untreated controls, consistent with the notion of increased symplastic connectivity.

The data fit the branch pattern predicted to arise in our model from an increase in the rate of global auxin transport, and provide a plausible mechanism by which auxin cytokinin and strigolactone could interact.

We have included these data as a new figure in the main text (Figure 9).

*It would be nice to have more comparison throughout the text and results sections with the classic decapitation experiments in angiosperms, what outcomes are the same? What is different? Likewise for the auxin/cytokinin/strigolactone up/down manipulations*.

We have substantially revised the manuscript and hope that this reviewer will now be satisfied.

Specific comments:

*In the subsection headed “Strigolactone fulfils a predicted requirement for basal suppression of branching”: If the basal inhibitor in the model is indeed strigolactone, then why doesn't a branch always occur in the basal most metamer in the* ppccd8 *mutant as in the model (compare*
Figure 9
*with*
Figure 3*)*?

In Arabidopsis, two carotenoid cleavage dixygenases MAX3/CCD7 and MAX4/CCD8 jointly regulate striglolactone biosynthesis. These roles are conserved in *Physcomitrella* and are carried out by *PpCCD7* and *PpCCD8*. Whilst *ppccd8* mutants have reduced strigolactone levels, they still have a partial strigolactone complement, and the expression pattern and developmental function of PpCCD7 is unknown. We have tried to clarify this issue in the main text.

*The phenotype looks pretty weak compared to model predictions, suggesting that strigolactone must not be the main player for the basal repression*.

We have shown that upregulating strigolactone biosynthesis is sufficient to suppress *Physcomitrella* branch initiation, and that branch initiation is specifically upregulated at the base of *ppccd8* mutant gametophores. These data implicate strigolactones as important contributors to basal branching.

Figure 3*: Why does the cell on the bottom, 9th from the right, look to have a gradient of auxin coming from it, as though it was an auxin source*?

The blue hexagons in Figure 3 represent the ratio between the auxin concentration (*c*) and the threshold for branch activation (*T*). In some simulated shoots in Figure 3, there is a high ratio of *c/T* because we have specified lower values of *T* in the basal metamers of the shoot. This means that the auxin concentration (*c*) does not drop below the threshold level for branch activation so readily. Therefore, there is no auxin emanating from the base.

We have tried to clarify these aspects of the model by changing the explanatory text and changing the visualization to better distinguish between consideration of the auxin concentration (*c*), and *c/T*. We have also altered the visualization to show metamers in which the basal inhibitory cue is active.

Reviewer #3:

*[…] Taken together the data support the model for branching. For the first time a coherent model that explains moss gametophore branching is presented*.

Suggestions:

*1) In the subsection headed “Auxin biosynthesis mutants capture predicted effects of changing model values H*_*apex*_
*and H”: This section is written cryptically. I don't have a problem with the data but the cryptic way in which the data are referred to. The authors should report the phenotype of* shi *mutants in the text. They only refer to H values, which is vague in the context of an experiment and experimental data. The final sentence of this section is unclear and vague—what is the biological result? The concluding sentence is no help to the reader (see helpful concluding sentence of the following paragraph). Even the title of*
Figure 6
*is unhelpful and gnomic*.

We have described the biological result in this sentence.

*2) In the subsection headed* “*Up-regulating cytokinin degradation captures predicted effects of decreasing model values of* T”*, the expression* “*similar to the model output*”*. It would help a struggling reader if the authors stated the result and then concluded that his was similar to the model output*.

We have revised the appropriate text to address these comments.